# Multi-point Dimensionality Reduction to Improve Projection Layout Reliability

## Abstract

In ordinary Dimensionality Reduction (DR), each data instance in a high dimensional space (original space), is mapped to one point in a low dimensional space (visual space). This builds a layout of projected points that attempts to preserve as much as possible some property of data such as distances, neighbourhood relationships, and/or topology structures, but with the ultimate goal of approximating semantic properties of data. The approximation of semantic properties, is achieved by preserving geometric properties or topology structures in visual space. In this paper, the first general algorithm of Multi-point Dimensionality Reduction is introduced on where each data instance can be mapped to possibly more than one point in visual space with the aim of improving reliability, usability and interpretability of dimensionality reduction. Furthermore, by allowing the points in visual space to be split into two layers while maintaining the possibility of having more than one projection per data instance, the benefit of separating more reliable points from less reliable points is discussed. The proposed algorithm in this paper, named Layered Vertex Splitting Data Embedding (LVSDE), is built upon and extends a combination of ordinary DR and graph drawing techniques. Based on the experiments of this paper, the particular proposed algorithm (LVSDE) practically outperforms popular ordinary DR methods visually in terms of semantics, group separation, subgroup detection or combinational group detection.

## 1  Introduction

Dimensionality Reduction (DR) or data embedding in a low dimensional space has gained widespread attention in different applications. Some DR techniques such as t-SNE Maaten & Hinton (2008) and UMAP McInnes et al. (2018a;b) have become essential tools in many application domains such as genomics, machine learning, NLP, cancer research, and protein folding. Typically, a DR technique maps data from a high dimensional space (original space), or on a distance matrix denoting original space distances, onto a two dimensional visual space, preserving some property of the data Nonato & Aupetit (2019).

DR techniques can be categorized into linear and non-linear. For linear DR techniques such as Principal Component Analysis (PCA) Jolliffe (1986) each dimension of visual space is a linear combination of dimensions of original space. But for non-linear DR techniques such as t-SNE Maaten & Hinton (2008) and UMAP McInnes et al. (2018a;b) the meaning of each dimension of the visual space is not as well-defined. Regardless, the possibility of errors in the resulting visual representations invites more attention as the presence of groups that do not exist, inaccurate neighbourhoods, mismatching topology structures, or mismatching distances are inevitable in many scenarios. Also semantically, in dimensionality reduction important information is sometimes lost or inadequately visualized to be properly perceivable.

To reduce those shortcomings and noting limitations in the ordinary definition of dimensionality reduction, this paper deeply explores a relaxed definition of dimensionality reduction called Multi-point Dimensionality Reduction (MDR) where each data instance can be mapped to multiple points in visual space. Then, this relaxation is augmented with layers in the definition of Multi-layered Multi-point Dimensionality Reduction (MMDR) where not only each data instance can be mapped to multiple points in visual space but also the set of projected points in visual space can be split into multiple layers.

The contributions of this paper can be categorized into philosophy discussion and empirical exploration. On the philosophy front, a clear picture of (Multi-layered) Multi-point Dimensionality Reduction as a general DR formulation is drawn by showing how and which problems it can solve either theoretically or practically from multiple perspectives. This discussion is built on top of previous efforts of others on related problems but the extent of discussion and general feasibility of MDR is unique to this paper. On the empirical side, it is shown that the particular proposed algorithm (LVSDE) practically outperforms popular single point DR methods visually by qualitative analysis. A quantitative analysis based on KNN classification is also provided to increase the confidence in the visual results.

The LVSDE algorithm proposed in this paper which is a tangible implementable instance of MMDR and therefore MDR, is studied on different data sets to show that it performs better than ordinary DR techniques visually with respect to semantics, group separation, subgroup detection or combinational group detection. To showcase these, different embeddings of different data sets consisting of genomic distances, handwritten digit images, flower plant features, email text features, and image classifier neural network layer outputs are illustrated in Figs. 1, 2, 10, 11, 12, 13, 14, 15, 16, 17 and 18.

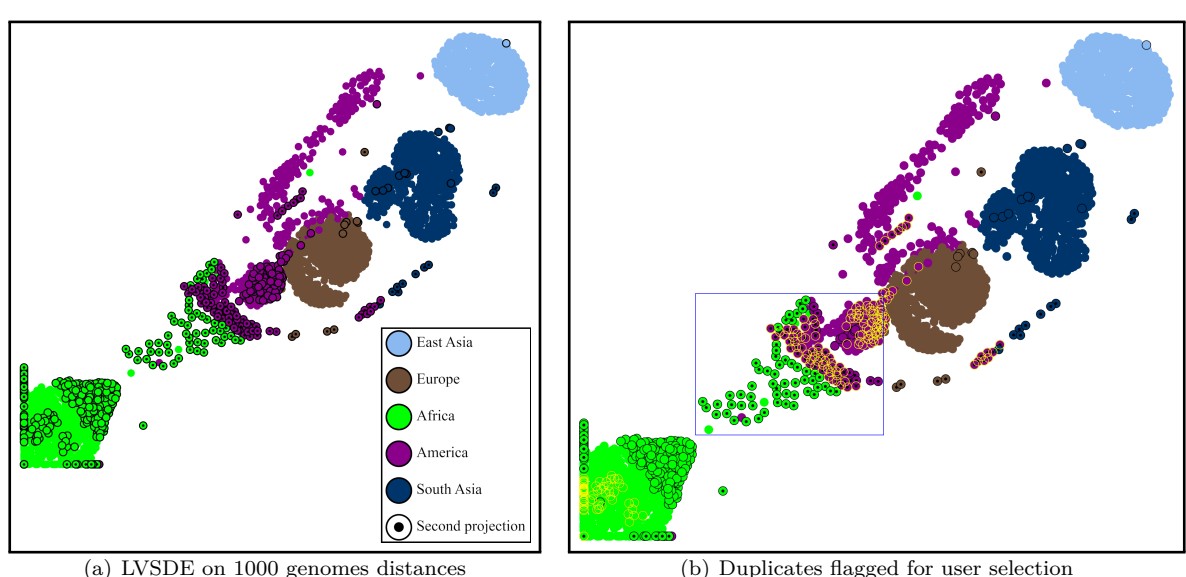

(a) LVSDE on 1000 genomes distances       (b) Duplicates flagged for user selection

Figure 1: (a) An LVSDE embedding of the 1000 genomes project data set Auton et al. (2015) distances. Points with a black circle around them are in the gray layer. Points with a black dot inside them are second projections of an input data instance. Points without a black circle around them are in the red layer. Duplication of some points in America from some area close to Europe into an area close to duplicates of some of points of Africa in an area of visual space between Europe and Africa, matches immigration from Europe and Africa to America. LVSDE has successfully selected points to duplicate and successfully guided them in a meaningful way. For LVSDE configuration 1 is used. (b) Yellow circles specify points that have a corresponding duplicate point in the blue rectangle meaning that for each point with yellow circle around it there is a point in the blue rectangle which is another projection of the same data instance of original space. The blue rectangle is specified by the user.

## 2 Related Work

While dimensionality reduction relies on preserving geometric properties and topology structures of data, the ultimate goal on the application front has been to visualize and group the semantic properties of data for that application. Such groupings are sometimes approximated with geometric properties and topology structures of data and then preserved into visual space.

An important shortcoming of dimensionality reduction with exactly one projection per data instance point is that it cannot reliably and to fully show independent belonging to multiple fuzzy sets. It is important to distinguish between the definition of fuzzy sets Zadeh (1965) and fuzzy clustering Ruspini (1969) as the definition of fuzzy clustering has an additional restriction that the fuzzy sets Zadeh (1965) definition does not have which is the sum of intensities of belonging to different clusters should be 1.

In the definition of fuzzy clustering Ruspini (1969) which has taken a more univariate approach, the intensity of belonging to each cluster depends on the intensity of belonging to the other clusters. But in the model of belonging to multiple fuzzy sets by Zadeh Zadeh (1965) there is no such restriction. In that model, the intensity of belonging to one cluster can change independent of the intensity of belonging to another cluster and is more in line with multivariate classification Anderson (1951) emerging as an important topic in many applications Morais et al. (2020); Liu et al. (2015); di Bella et al. (2013).

While ordinary dimensionality reduction has been used in a user study for exploring fuzzy clusterings by Ying Zhao, et al. Zhao et al. (2019), it can be only accurate or reliable for univariate fuzziness in fuzzy clustering Ruspini (1969) rather than the multivariate model of belonging to multiple fuzzy sets Zadeh (1965). Also, our proposed method is examined on more realistic data that has not already gone through a fuzzy clustering algorithm. Moreover, our method goes beyond univariate fuzzy semantics and visualizes multivariate fuzzy semantics more accurately and more reliably.

A single point shown between two groups of points in visual space can indicate some form of fuzzy relationship to the two groups but not every form of fuzzy relationship. If a point is between two groups of points, then moving toward one group will be moving away from the other group. This restricts the possibility of independently specifying the intensity of belonging to the first group and the intensity of belonging to the second group using any position-based visual clue. Having possibly more than one projection per data instance can be a better indicator of unrestricted belonging to multiple fuzzy sets by allowing different indicators for each fuzzy set. The existence of such indicators improves the potential of preserving geometric and topology structures of data for approximating multivariate fuzzy semantics.

In visual space, the Euclidean distance from an area of visual space, is generally used as a visual clue for how related a point is. But the fact that Euclidean distance has to adhere to the metric triangular inequality, restricts the closeness it can imply to different areas of visual space. This creates a mismatching model if the closeness relationship in the data to be visualized does not adhere to the triangular inequality.

The inability of ordinary dimensionality reduction to properly visualize non-metric distances is well highlighted by Van der Maaten and Hinton Van der Maaten & Hinton (2012) and Cook et al. Cook et al. (2007). However, they fell short of offering a comprehensive solution to the problem they raised. Their idea of embedding into to multiple maps, each with exactly one projection per data instance, does not adequately resolve the problem and causes more issues than it fixes. One reason that their idea causes issues is the effect of change blindness Simons & Ambinder (2005); Simons & Levin (1997) on multiple embeddings. Another reason is the observation that the aggregate sum of the total number of projected points in all of the multiple embedding maps is unjustifiably high. Indeed, embedding into a single map with possibly more than one projection per data instance is a more succinct and direct approach.

It is so because of the aforementioned reasons and also because the number of projected points needed to explain the same set of relationships between projected points is lower. Furthermore, more relationships are visualized in a single view giving the viewer a better chance to reason on them.

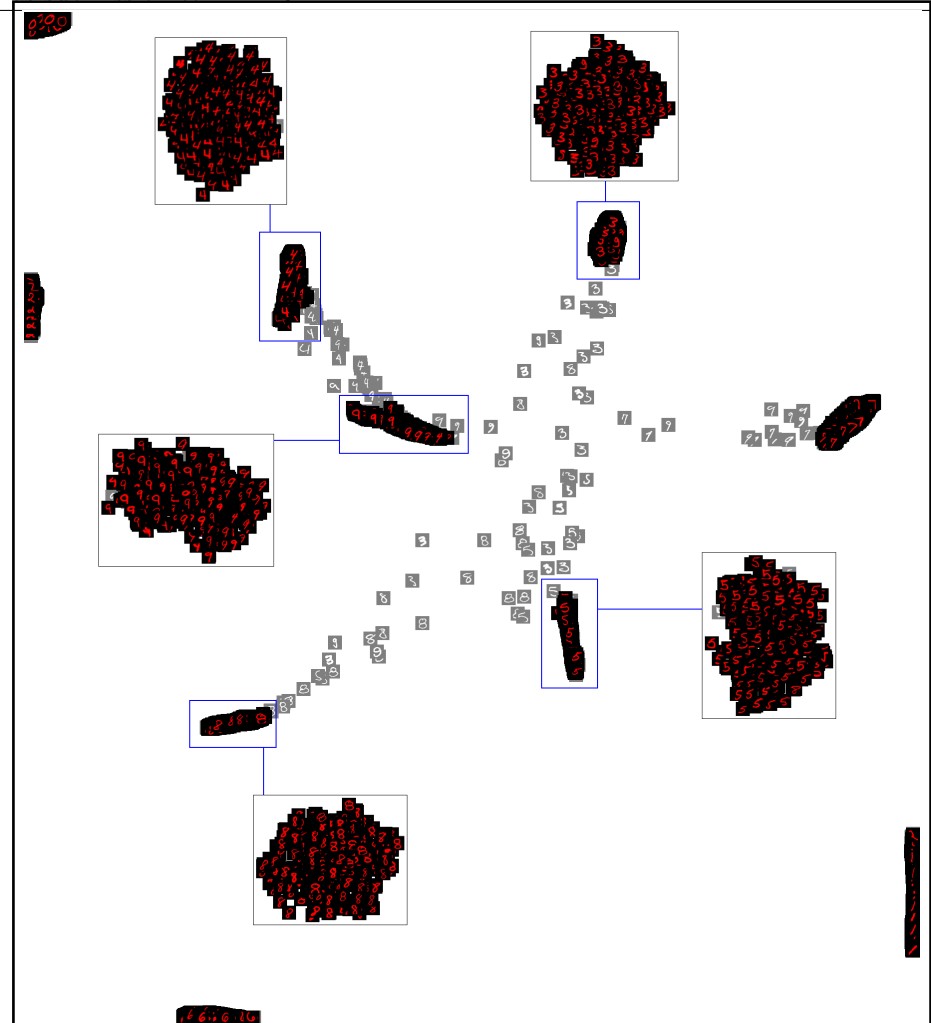

(a) LVSDE on a subset of MNIST in addition to overlap reduction on manually selected regions

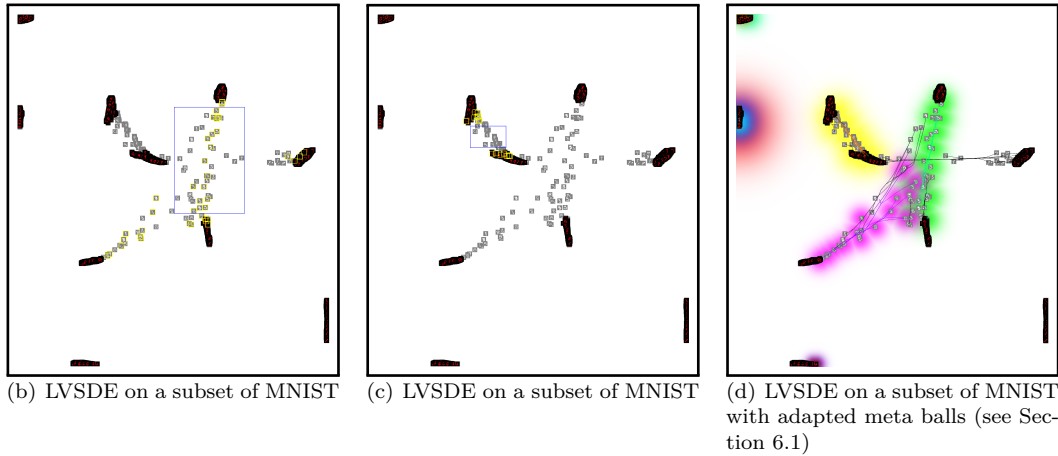

(b) LVSDE on a subset of MNIST    (c) LVSDE on a subset of MNIST    (d) LVSDE on a subset of MNIST with adapted meta balls (see Section 6.1)

Figure 2: (a) An LVSDE embedding of a subset of MNIST data set using the information from the whole data set training data. A variant of overlap reduction algorithm of Nachmanson et al. (2016) by Nachmanson et al. is used on the points in blue rectangles to achieve the layouts in the attached rectangles. Separation of digits 3,5 and 8 on the red layer are better than popular techniques. Separation of digits 4 and 9 on the red layer are better than popular techniques. By using layers, the identification of groups is much easier when only considering red layer while gray layer provides some background information. (b,c) Yellow rectangles specify points that have a corresponding duplicate point in the blue rectangle. The blue rectangle is specified by the user.

An interesting argument emerges when looking at the fact that for many applications there exists data already in a non-metric distance form. If the semantic distance in data can be non-metric, then why is Euclidean distance usually used to model the semantics of neighbourhoods in the original space, knowing that Euclidean distance is a metric distance? One might argue that Cosine distance has been used too and Cosine distance is not a metric distance. But Cosine distance still is far from an unrestricted distance model as it equates to half of squared Euclidean distance of normalized vectors.

Borrowing some ideas from Principal Component Analysis (PCA) Jolliffe (1986), this paper argues that semantic neighbourhoods should be modelled on the set of perpendicular geometric projections of points of original space on each hyper-line of a specific set of hyper-lines or each manifold of a specific set of manifolds[1]. The specific hyper-lines or manifolds are meant to represent different semantic aspects of data since assuming just one aspect for data is an oversimplification of an important problem. Then a point can have several semantic neighbourhoods defined around it based on different hyper-lines or manifolds. The challenge becomes how to bring those neighbourhoods to a visual space usually perceived through Euclidean distance which has to adhere to the triangular inequality.

Multi-point Dimensionality Reduction (MDR) can better show multiple neighbourhoods around a single point of original space by having multiple projections of it in visual space. This is because MDR gives each data instance more freedom to express itself in different neighbourhoods of visual space using multiple projected points of it. Also, layers can improve the interpretation of those neighbourhoods of original space in visual space by the separation of the set of visualized points. A simplification of multiple neighbourhoods around a point of original space can be understood from Fig. 3 example.

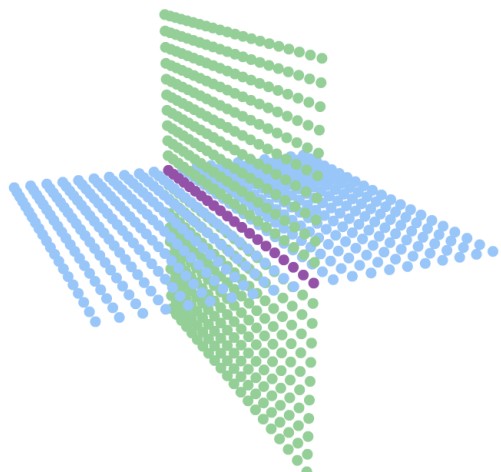

Figure 3: Points on the surface of two planes regularly distributed. The points on the purple line have two distinct neighbourhoods based on two different planes.

Visualizing multiple neighbourhoods has also recently been the focus of work of Huroyan, et al. Huroyan et al. (2022) where they use reduction to three dimensions as a remedy for simultaneous neighbourhoods. However, it has been long disputed whether 3D visualizations rendered on a 2D display limits the ability of perceiving all information. Nevertheless, in Barahimi & Wismath Barahimi and Wismath argued that the virtual reality technologies and stereoscopic 3D might provide some improvement over monocular 3D though this would represent significant hardware commitment.

Specifically, for word embeddings, research by Huang et al. Huang et al. (2012) about studied how different contexts should affect them. While their technique is specific to word embeddings and is not generalizable to other applications, it also requires additional context data, distancing it to some extent from the dimen-

---

[1]The meaning of perpendicular geometric projection on a manifold, is based on the normal vector of the tangent space on any point on the manifold creating a perpendicular ray from any point on the manifold which means a point can have multiple perpendicular geometric projections on a manifold.

sionality reduction paradigm. In this paper, not only is a general solution (LVSDE) proposed applicable to different application domains, it does not require any context data and tries to find the semantic contexts implicit in the data.

Although preservation of a well-defined topology structure is not the main objective of this paper, we partially address the challenge of visually connecting multiple projections of a point which are notable steps forward for this important issue. To go from a finite set of points in a high dimensional space or even a distance matrix to a topological structure, one known method is Vietoris-Rips filtration Bauer (2021); Edelsbrunner & Harer (2010) sometimes simply referred to as Rips filtration. Persistence diagrams are a way to represent such topological structures. Relevant research in preserving topology structures, is the interesting prior work of Doraiswamy et al.Doraiswamy et al. (2020) in which they preserve 0-dimensional persistence diagram of the Rips filtration Bauer (2021); Edelsbrunner & Harer (2010) of high dimensional data.

AupetitAupetit (2007), studied the question of whether, as a result of dimensionality reduction a manifold in original space that data instances were laying on is torn in visual space. While the study stops short of giving any theoretical or empirical remedy to preserve neighbourhoods when tearing happens, it is related to the theoretical argument made in Section 3 regarding if MMDR can maintain neighbourhoods when manifold tearing happens. For the particular example of points on a 3D cylinder that is discussed in Section 3, no empirical solution is provided in this paper and rather the discussion is only framed in a theoretical fashion.

While defining neighbourhoods can be symmetric, asymmetric or partially symmetric, Van der Maaten and Hinton Maaten & Hinton (2008) opted for symmetry in their t-SNE paper Maaten & Hinton (2008) where they described the addition of symmetry as one of the major differences of t-SNE from its predecessor SNE Hinton & Roweis (2002). The desirability of symmetry of neighbourhoods is also reflected in our proposed algorithm in its neighbourhood normalization transformation. However the move is toward more symmetry but not necessarily total symmetry because if the nature of data has asymmetric neighbourhoods, making the neighbourhoods totally symmetric may remove one of the important structures of original space data in visual space which is in opposition to the goal of preserving structures. As dimensionality reduction reduces the dimensionality of data, it also makes sense to reduce but not remove the asymmetry of data if it has one (symmetry can be usually preserved without a need for reduction but asymmetry is easier to hold in higher dimensions than lower dimensions).

## 3    Problem Formalization and Visual Metaphors

To formalize the problem, let's first recall the definition of ordinary dimensionality reduction (ODR) or data embedding. We are given a set of data instances $Q = \{q_1, q_2, ..., q_n\}$ either specified by points in an $m$-dimensional space called original space where each $q_i \in \mathbb{R}^m$, or specified on rows and columns of a matrix $\bar{\delta}$ denoting original space distances. Then the goal is to project each $q_i$ onto a point $p_i$ in a low dimensional space called visual space (usually 2-dimensional) so that some of geometric properties or topology structures, are preserved as much as possible Martins et al. (2014); Lee & Verleysen (2007).

One approach to dimensionality reduction, is the distance preservation strategy. Let's assume distances in the original space between $q_i$ and $q_j$ are denoted by $\bar{\delta}(q_i, q_j)$ and distances in the visual space between $p_i$ and $p_j$ are denoted by $D_v(p_i, p_j)$. The strategy tries to minimize the difference between $D_v(p_i, p_j)$ and $\bar{\delta}(q_i, q_j)$ for all $1 \leq i < j \leq n$ or some variant of that Nonato & Aupetit (2019).

Although well-accepted, if distances are Euclidean, this minimization usually fails to reach zero difference for all $1 \leq i < j \leq n$, even for simple 3D data sets, such as points on a cylinder's surface as shown in Figure 4(a). The failure expands to preserving local neighbourhoods too which is better addressed with duplicate points as shown in Figure 4(b). Duplicate points that are not allowed in ODR, but are allowed in the MDR defined bellow, enable a data instance to be projected onto two distant neighbourhoods in visual space. This is done by allowing two separate projections for the same data instance.

**Definition 3.1.** *Multi-point Dimensionality Reduction.*

*Given a set of data instances $Q = \{q_1, q_2, ..., q_n\}$ either specified by points in an m-dimensional space called original space where each $q_i \in \mathbb{R}^m$, or specified on rows and columns of a distance matrix denoting original*

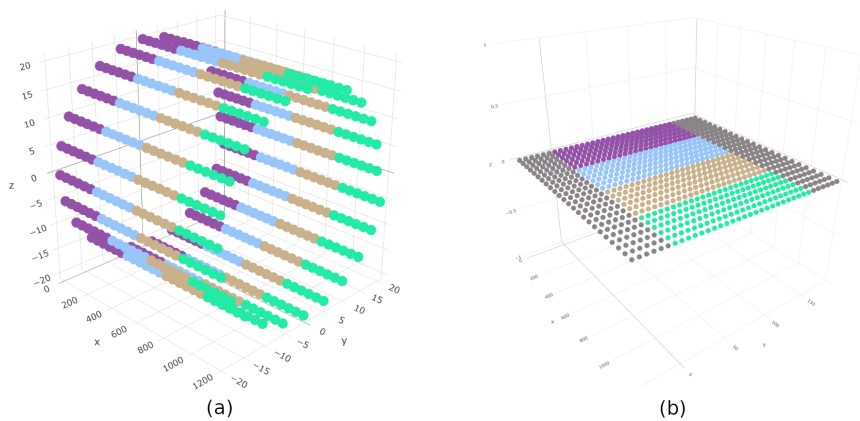

Figure 4: (a) Points on the surface of a cylinder regularly distributed. (b) The torn and unfolded cylinder with duplicate points added to the sides where the points coloured gray on each side are duplicates of the points with non-gray colour on the other side (the image is not the result of any particular empirical dimensionality reduction method and only highlights the potential of Multi-layered Multi-point Dimensionality Reduction as a problem formulation in a theoretical fashion).

space distances, an MDR maps each data instance $q_i$ to a non-empty set of points $S_i = \{s_{i_1}, s_{i_2}, ..., s_{i_{\lambda_i}}\}$ where $\forall s_{i_t} \in \mathbb{R}^2$.

With this definition, a variant of the classical distance preservation problem for ordinary dimensionality reduction can be transformed to MDR as follows:

**Definition 3.2.** *Multi-point Dimensionality Reduction Distance Preservation Problem.*

*The MDR Distance Preservation Problem seeks to find an MDR that minimizes*

$$\frac{\sum_{i=1}^{n} \sum_{j=1}^{n} (\min_{(s_{i_t}, s_{j_r}) \in S_i \times S_j} ([\bar{\delta}(q_i, q_j) - D_v(s_{i_t}, s_{j_r})]^2))}{n^2} \tag{1}$$

*subject to $\sum_{i=1}^{n} |S_i| \leq \varphi$, where $\varphi$ is a constant for limiting the number of points in visual space.*

In Section 2, the benefit of splitting the set of projected points into different layers for distinction between different neighbourhoods around a projected point was discussed. By combining the idea of layers with the idea of more than one projection per data instance, Multi-layered Multi-point Dimensionality Reduction (MMDR) is defined as follows:

**Definition 3.3.** *Multi-layered Multi-point Dimensionality Reduction.*

*Given a set of data instances $Q = \{q_1, q_2, ..., q_n\}$ either specified by points in an m-dimensional space called original space where each $q_i \in \mathbb{R}^m$, or specified on rows and columns of a distance matrix denoting original space distances, an MMDR maps each data instance $q_i$ to a non-empty set of points $S_i = \{s_{i_1}, s_{i_2}, ..., s_{i_{\lambda_i}}\}$ where $\forall s_{i_t} \in \mathbb{R}^2$, and also maps each $s_{i_t} \in S_i$ to a layer number $\eta(s_{i_t})$. For convenience $\gamma_{i_\theta} \subseteq S_i$ is defined as the subset of points in $S_i$ such as $s_{i_\Omega}$ for which the condition $\eta(s_{i_\Omega}) = \theta$ holds, meaning that $\gamma_{i_\theta}$ is the set of projections of $q_i$ in the layer $\theta$ of visual space.*

One of the challenges introduced by MMDR is that a point might not have any projection in one of the layers in visual space. If only that layer is displayed, this can hide some relevant information. One approach to address this challenge is to display all layers but with a different visual indicator for each layer such as colour. While the interpretation of different layers is open in the definition of MMDR, this paper emphasizes a particular type of MMDR where there are only two layers named red layer and gray layer. The red layer is intended for more reliable points and the gray layer is intended for less reliable points. As a result, a Red Gray Embedding is defined below:

**Definition 3.4.** *Red Gray Embedding.*

*A Red Gray Embedding, is an MMDR with just two layers named red layer and gray layer. The red layer is intended for more reliable points and the gray layer is intended for less reliable points.*

In the above definition a point can have several projections in the red layer and several projections in the gray layer at the same time. The following stricter definition is more pragmatic as it is explained after.

**Definition 3.5.** *Strict Red Gray Embedding.*

*A Strict Red Gray Embedding, is an MMDR with just two layers named red layer and gray layer where points that have more than one projection are mapped to the gray layer (points that have exactly one projection can be mapped to either red layer or gray layer). The red layer is intended for more reliable points and the gray layer is intended for less reliable points.*

Such a strict definition is based on experimental results that if a point is projected unreliably, most likely it is because it is related to multiple groups of points as independent separate fuzzy sets in a way that is not consistent with triangular inequality. Therefore having more than one projection might help mitigate the unreliability.

## 4 Method

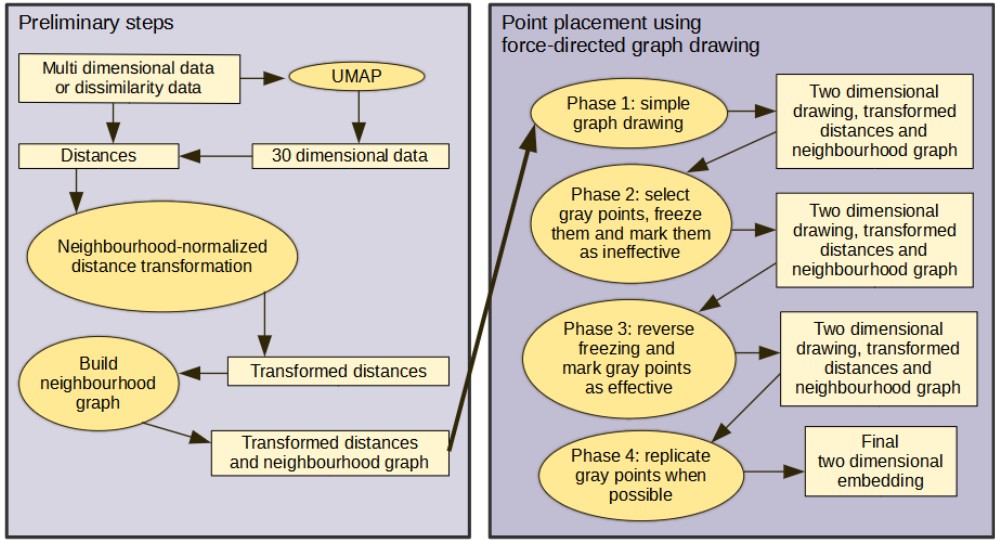

Figure 5: Summary of the proposed dimensionality reduction algorithm of this paper (LVSDE) showing the flow of data through different components of the algorithm

In this section, the proposed algorithm of this paper named Layered Vertex Splitting Data Embedding (LVSDE) is discussed which produces a Strict Red Gray Embedding. In this discussion $D_v$ indicates distances in visual space, $\bar{\delta}$ for distances in original space before any transformation, and $\delta$ for original space distances after transformation using the neighbour-normalized distance transformation. Also $D_{v_{max}}$ is used to denote the maximum distance in visual space and $\delta_{max}$ is used to denote the maximum of transformed distance of original space.

### 4.1 Overview

An overall of the proposed algorithm (LVSDE) is summarized in Fig. 5. The algorithm builds upon, extends and modifies a combination of UMAP McInnes et al. (2018a;b), the Fruchterman and Reingold Fruchterman & Reingold (1991) force-directed graph drawing algorithm and the force scheme Tejada et al. (2003) embedding technique.

The proposed algorithm in this paper has three preliminary steps and four phases. The first preliminary step which is UMAP McInnes et al. (2018a;b) to 30 dimensions, is optional as it is discussed later. The second preliminary step is the neighbourhood-normalized distance transformation. The third preliminary step is to build a neighbourhood graph $G$ based on transformed distances from the previous preliminary step. Once a neighbourhood graph is ready a four phase force-directed graph drawing process is started.

While Section 4.2 elaborates on the three preliminary steps of the proposed algorithm, the Section 4.3 which follows it, elaborates the four phases through a force-directed graph drawing.

## 4.2 Preliminary Steps

### 4.2.1 UMAP to 30 dimensions

The first preliminary step which is UMAP to 30 dimensions, is optional. The reason for this is to benefit from its manifold learning capabilities when sufficient portion of the data is uniformly distributed on a Riemannian manifold and to be able to skip it when that is not the case. In some scenarios, another benefit is that UMAP is much faster than the rest of the proposed algorithm and therefore such a preliminary step allows partially learning the embedding from a larger data size and the rest of the algorithm uses a smaller data size.

If the user opts to use UMAP to 30 dimensions as a preliminary step, then a choice needs to be made on how to compute distances on the resulting 30 dimensional data. While the typical choice here is Euclidean distance, another possibility (especially for text data sets) is Cosine distance and those distances will be considered as original space distances ($\bar{\delta}$) for the rest of the algorithm.

### 4.2.2 Neighbourhood-normalized distance transformation

The second preliminary step is neighbourhood-normalized distance transformation. This algorithm relies on drawing a $\hat{p}$ nearest neighbour graph where $\hat{p}$ is a constant parameter indicating neighbourhood size. But nearest neighbour graphs with a constant number of neighbours for each vertex are directed graphs, and the graph drawing algorithm chosen to be modified is designed for undirected graphs. The distance transformation named *neighbourhood-normalized distance transformation* that is introduced below at the very least is designed to make neighbourhood graphs more balanced where more balanced means fewer edges without an edge in the opposite direction, and in practice is very effective.

Given a neighbourhood size $z$ (default value $z = 20$), for each point $q_i$ in $Q$ a constant value $m_i$ is computed such that:

$$tan^{-1}(\bar{\delta}_{i_z} \cdot m_i) = 1$$

where $\bar{\delta}_{i_z}$ is the Euclidean distance or Cosine distance or a custom precomputed distance from the $z^{th}$ nearest neighbour of $q_i$ in the original space to $q_i$. The *neighbourhood-normalized distance* is then defined as:

$$\delta(q_i, q_j) = \frac{tan^{-1}(\bar{\delta}(q_i, q_j) \cdot m_i) + tan^{-1}(\bar{\delta}(q_i, q_j) \cdot m_j)}{2} \tag{2}$$

where $\delta(q_i, q_j)$ becomes the transformed distance between $q_i$ and $q_j$, and $\bar{\delta}(q_i, q_j)$ is the Euclidean distance or Cosine distance or a custom precomputed distance between $q_i$ and $q_j$. The repeating of $\bar{\delta}(q_i, q_j)$ in Equation 2 is not a typo as discussed later and the custom precomputed distance mentioned above corresponds to the case where the input data is in distance matrix form.

Fig. 6 is helpful in understanding the computation of $m_i$ for the neighbourhood-normalized distance transformation. The density of distances around each data instance of original space is brought into play by $\bar{\delta}_{i_z}$ in a similar way to t-SNE computes and uses a per data instance standard deviation $\sigma_i$ of Gaussian distribution in order to bring the density of distances around each data instance of original space into play. The role that the value $z$ plays is in analogy with perplexity for t-SNE. Needless to say, t-SNE being a statistical and probabilistic method, uses a different paradigm than the geometric neighbourhood-based graph drawing paradigm that is used in this paper but the requirements of a good embedding remain the same.

One important observation to make is that in a nearest neighbours graph with a constant number of neighbours for each vertex, an edge without an edge in the opposite direction between the same pair of vertices, is an indication of density change. So in a more balanced neighbourhood graph, then in some regions, either density changes become smoother or a big separation replaces them. This depends on how the conflict of an edge without edge in the opposite direction is resolved for some edges to either two edges in the opposite direction or no edge at all.

In most cases the original space distances before any transformation are symmetric, and the distinction between $\bar{\delta}(q_i, q_j)$ and $\bar{\delta}(q_j, q_i)$ are not relevant to Equation 2. In rare cases where the original space distances before transformation might be non-symmetric, if $\bar{\delta}(q_j, q_i) \cdot m_j$ was used instead of $\bar{\delta}(q_i, q_j) \cdot m_j$ in Equation 2, it would transform a non-symmetric distance matrix into a symmetric distance matrix, removing an important property of input data. However, the distance transformation still brings non-symmetric distances closer to symmetric but not to completely symmetric.

The reason that the proposed method does not synchronize $z$ (the neighbourhood size of neighbourhood-normalized distance transformation) with the neighbourhood size for building the neighbourhood graph ($\widehat{p}$), is that a change to $\widehat{p}$ would have brought a more dramatic change to the final embedding. However, the objective was to reduce the volatility of parameters.

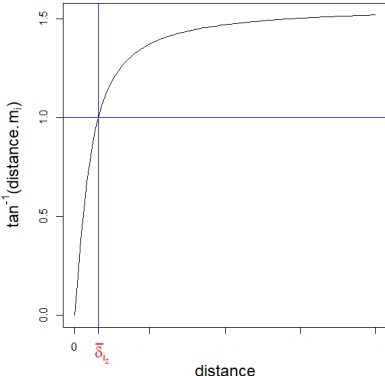

Figure 6: Computing $m_i$ for neighbourhood-normalized distance transformation

### 4.2.3   Building the neighbourhood graph

The third preliminary step is to build a neighbourhood graph $G$ based on transformed distances from the previous preliminary step choosing the $\widehat{p}$ nearest points as the neighbours of a point, where $\widehat{p}$ is a constant parameter specifying the number of neighbours for each point. The lower that $\widehat{p}$ is chosen, the more local the method becomes.

## 4.3   Point Placement Using Force-Directed Graph Drawing

In the four phases of the proposed algorithm, force-directed graph drawing Fruchterman & Reingold (1991) is used but several modifications are made.

### 4.3.1   Brief review of the Fruchterman and Reingold Fruchterman & Reingold (1991) algorithm

The algorithm in Fruchterman & Reingold (1991) for drawing a graph in two dimensions, uses two different sets of forces based on the geometric distance in the drawing and adjacency of vertices. The first set of forces is called repulsive forces and is applied to every pair of graph vertices. The second set of forces is called attractive forces and is applied only to adjacent vertices. The forces are applied to vertices through multiple iterations. This assumes a unit time (in physics) between two iterations, unit mass for each vertex, and a

constant speed during the iteration only changed from zero by acceleration at the beginning of the iteration. The movement of vertices by forces is limited to a gradually decreasing temperature. Temperature can be seen as the radius of a circle around the location of a vertex in the previous iteration. The vertex is stopped on the edge of the circle in the current iteration if it tries to move outside the circle, limiting the amount of movement in each iteration for each vertex.

The repulsive force $(\vec{f_r})$ is imposed by any unordered pair of vertices on each of the two vertices and the attractive force $(\vec{f_a})$ is imposed by each edge on each of its endpoints as they were defined in Fruchterman & Reingold (1991). But when brought into the conceptual context and notation of this paper, they are in the form of:

$$\vec{f_r} = -\frac{\Gamma^2}{D_v(s_{i_t}, s_{j_r})} \cdot \vec{\alpha}$$

and

$$\vec{f_a} = \frac{(D_v(s_{i_t}, s_{j_r}))^2}{\Gamma} \cdot \vec{\alpha}$$

where

$$\vec{\alpha} = \begin{cases} \frac{s_{j_r} - s_{i_t}}{|s_{j_r} - s_{i_t}|} & \text{if the force is applied on } s_{i_t} \\ \\ \frac{s_{i_t} - s_{j_r}}{|s_{i_t} - s_{j_r}|} & \text{if the force is applied on } s_{j_r} \end{cases} \tag{3}$$

In the notation used, $s_{i_t}$ and $s_{j_r}$ are projections of a data instance in visual space and therefore a vertex of the neighbourhood graph, and $\Gamma$ is a constant optimal distance defined as:

$$\Gamma = \sqrt{\frac{width \times height}{NumberOfPoints}}$$

However, while it would be natural to assume the $NumberOfPoints$ in the definition of $\Gamma$ refers to the number of projected points in visual space, but, that would not be constant for the proposed algorithm of this paper. So instead a definition of $\Gamma$ is opted for where $NumberOfPoints$ refers to number of data instances in original space ($|Q|$) which is a constant, so the definition of $\Gamma$ becomes:

$$\Gamma = \sqrt{\frac{width \times height}{|Q|}} \tag{4}$$

Moreover, whether edges of a neighbourhood graph should be considered in a directed fashion or not is determined by considering edges as directed and if there are two edges in opposite directions between two vertices, we consider them separate edges and therefore apply separate forces for them. To decrease the temperature the simplest approach (as mentioned by Fruchterman & Reingold (1991)) is to start with an initial temperature such as one tenth of the width of drawing and linearly decrease temperature so that it becomes zero in the last iteration. The maximum temperature is denoted by $\bar{u}$ and the default value of $\bar{u} = 100$ is used in the implementation.

### 4.3.2 Modifications to the computation of forces and usage of masses in acceleration

Although the graph layout algorithm of Fruchterman & Reingold (1991) is a good approach, it is not designed for the purpose of dimensionality reduction, so some modifications are proposed in this paper for that purpose.

Our first modification is to change the attractive forces for any edge $(s_{i_t}, s_{j_r})$ of the neighbourhood graph $G$ on $s_{i_t}$ or $s_{j_r}$ to the form:

$$\vec{f_a} = \psi \cdot \vec{\alpha} \tag{5}$$

where

$$\psi = \left( \frac{D_v(s_{i_t}, s_{j_r})}{\Gamma} \right)^{(1-b)} \tag{6}$$

and where the parameter $b$ is a *visual density adjustment parameter* (default $b = 0.9$).

This modification, changes the balance between attractive and repulsive forces in a way that is in contrast with two characteristics of the Fruchterman & Reingold (1991) algorithm: distributing vertices evenly and uniform edge lengths. Although distributing vertices evenly and uniform edge lengths are considered good properties in general graphs, they are not good properties for dimensionality reduction, mainly because they can hide the inherent structure in the original space. While without this change the forces would attempt to achieve a uniform density connected visual space, this change can split the visual space into different sections as attractive forces can become more dense in one part than another.

In particular the parameter $b$, controls how much the density of edges in the induced subgraph of any subset of vertices of the neighbourhood graph should be reflected in the density of projected points (vertices) in the visual space. The induced subgraph can be any induced subgraph of $G$. If the vertices of the induced subgraph correspond to a semantic group for an application of dimensionality reduction, and if the density for the part of visual space corresponding to those vertices is different from the density of surrounding area, a separate visual group is perceived for that semantic group. This argument is more indicative of necessity than sufficiency. Sufficiency of the argument is studied empirically on various data sets and reflected in the results of the experiments of this paper.

To find a good value for $b$ quantitative metrics, the number of projected points on the frame border, the type of data set, size of data set, or visual quality may be used. But quantitative metrics that do not rely on classes, or the number of projected points on the frame border, may be more practical when dealing with data in which the classes are not known or the type of data set is not familiar.

The second modification that is proposed in this algorithm is to allow to some degree the exact value of distances in the original space (after transformation) to affect the forces in the visual space already defined based on the neighbourhood graph, the distances in the visual space and $\Gamma$. The reason is that although the neighbourhood graph is built on the transformed distances it does not retain the exact value of transformed distances and therefore is blind to the relative importance of neighbours. So a multi-objective approach is pursued here were the embedding of the neighbourhood graph has a higher priority but relative distance preservation can fine tune the embedding. Therefore in the second modification, attractive forces are modified again to the form:

$$\vec{f_a} = \widehat{\psi} \cdot \vec{\alpha}$$

where

$$\widehat{\psi} = \psi + \begin{cases} \min\left( \frac{|\psi|}{2}, h \right) & \text{if } h > 0 \\[2ex] \max\left( \frac{|\psi|}{-2}, h \right) & \text{if } h \leq 0 \end{cases}$$

where $h$ is a force defined based on distances in the original space (after transformation) and $h$ is in the form $h = \frac{\delta(q_i, q_j)}{\delta_{max}} - \frac{D_v(s_{i_t}, s_{j_r})}{D_{v_{max}}}$. The definition of $h$ borrows some ideas from the force scheme algorithm of Tejada et al. (2003) where distances of original space are the core of force computations. In the implementation of this algorithm, $D_{v_{max}}$ is only computed once after the initial random layout and used for subsequent iterations as well.

It is important to note that in the second modification, the effect of $h$ on the attractive force is limited to only half of the attractive force obtained in the previous modification, in order to stop it from dramatically changing the embedding. While the neighbourhood graph creates the substance of the embedding, exact values of transformed distances help fine tune the embedding in order to retain the significance of effect of the neighbourhood graph.

A third relevant modification, but not to the forces, is that the proposed algorithm in this paper opts to start from a unit mass for each vertex (projected point) but changes it when duplicating a vertex. This is

done to account for the lower number of edges on a vertex in the process of duplicating that is discussed later in this paper. In contrast, the algorithm of Fruchterman & Reingold (1991) conceptually assumes unit mass for each vertex for computing acceleration of a vertex from aggregate forces on a vertex, when calculating acceleration by attractive forces[2]. In this paper, the mass for each projected point $s_{i_t}$ denoted as $\omega_{i_t}$ is initially set to 1.

For acceleration of a vertex by a repulsive force $\vec{f_r}$, a unit mass is assumed for each vertex because the repulsive forces are not based on the edges of the neighbourhood graph and therefore the changes made to the neighbourhood graph is not relevant to them.

While Procedure 1 summarizes how repulsive forces are computed and applied to projected points on each iteration, Procedure 2 summarizes how attractive forces are computed and applied to projected points on each iteration.

---

**for** $i \in \{1, 2, ..., |Q|\}$ **do**
    **for** $j \in \{1, 2, ..., |Q|\}$ **do**
        **for** $s_{i_t} \in S_i$ **do**
            $T_{i_t} = (0, 0)$    Comment: $T_{i_t}$ is a 2D vector.
            **for** $s_{j_r} \in S_j$ *and* $s_{j_r} \neq s_{i_t}$ **do**
                **if** $s_{i_t}$ *is not frozen and* $s_{j_r}$ *is not ineffective* **then**
                    $T_{i_t} = T_{i_t} - \frac{\Gamma^2 \cdot (s_{j_r} - s_{i_t})}{|s_{j_r} - s_{i_t}|^2}$
                **end if**
            **end for**
            **if** $|T_{i_t}| \leq temperature$ **then**
                $s_{i_t} = s_{i_t} + T_{i_t}$
            **else**
               $s_{i_t} = s_{i_t} + \frac{T_{i_t}}{|T_{i_t}|} \cdot temperature$
            **end if**
            **if** *there is a border frame and* $s_{i_t}$ *is outside the frame* **then**
                Bring $s_{i_t}$ on the frame.
            **end if**
         **end for**
    **end for**
**end for**

**Procedure 1:** Summarizes how repulsive forces are computed and applied to projected points on each iteration.

### 4.3.3 Computing replication pressure in phase 2 and modifying the neighbourhood graph in phase 4

In phase 2 to be described in section 4.3.6 and phase 4 to be described in section 4.3.8, in each iteration a replication pressure per projected point is computed which is a measure how misplaced a projected point is in the layout of the embedding. To compute the pressure per point based on the attractive and repulsive forces on that point, a circle centered at that point is considered to form 36 axis. The radius of the circle is arbitrary. The perimeter of the circle is equally divided to 36 segments (arcs). The 36 different axes on the projected point are formed by having the projected point as the origin of the axis, and the midpoint of one of the segments incident on the positive side of the axis. This way, each of the 36 axis is radially $10°$ apart from the previous one starting with a horizontal axis on the projected point and the origin of all of the axis is the projected point. A conceptual picture of the 36 axis around a projected point is depicted in Fig. 7. Here 36 specifies the precision of duplication process and the precision of choosing the gray layer projected points.

For each axis, all force vectors on that projected point are first perpendicularly projected onto that axis and the sum of magnitude of projected vectors that are on the positive side of the axis, is considered as the positive pressure on that axis. Also, the sum of magnitude of projected vectors on the negative side of the axis is considered as the negative pressure. The replication pressure of the axis on the projected point is defined as the aggregate sum of the positive and the negative pressure on that axis. The replication pressure of the projected point is then computed as the maximum of replication pressures among all of the 36 axis on the projected point.

---

[2]The physics concepts used to describe the algorithm of Fruchterman & Reingold (1991) are interpreted from their formulas and not necessarily reflected in their paper as physics concepts

In phase 4 to be described in section 4.3.8, when trying to duplicate a projected point of the gray layer, the neighbourhood graph is modified based on the axis which produced the maximum replication pressure. When a projected point is duplicated, the projected point that was duplicated is denoted as the first point of duplication and the projected point that was added is denoted as the second point of duplication. This means that the second one becomes another mapping of the same data instance initially located at the same location in visual space.

---

**for** $i \in \{1, 2, ..., |Q|\}$ **do**
    **for** $s_{i_t} \in S_i$ **do**
        $T_{i_t} = (0, 0)$      Comment: $T_{i_t}$ is a 2D vector.
    **end for**
**end for**
**for** $i \in \{1, 2, ..., |Q|\}$ **do**
    **for** $s_{i_t} \in S_i$ **do**
        **for** $s_{j_r}$ *in neighbors of* $s_{i_t}$ *in G* **do**
            **if** $s_{i_t}$ *is not frozen and* $s_{j_r}$ *is not ineffective* **then**

$$\psi = \left( \frac{D_v(s_{i_t}, s_{j_r})}{\Gamma} \right)^{(1-b)}$$

$$h = \frac{\delta(q_i, q_j)}{\delta_{max}} - \frac{D_v(s_{i_t}, s_{j_r})}{D_{v_{max}}}$$

                **if** $h > 0$ **then**
                    $\widehat{\psi} = \psi + \min(\frac{|\psi|}{2}, h)$
                **else**
                    $\widehat{\psi} = \psi + \max(\frac{-|\psi|}{2}, h)$
                **end if**

$$T_{i_t} = T_{i_t} + \frac{\widehat{\psi} \cdot (s_{j_r} - s_{i_t})}{\omega_{i_t} \cdot |s_{j_r} - s_{i_t}|}$$

$$T_{j_r} = T_{j_r} + \frac{\widehat{\psi} \cdot (s_{i_t} - s_{j_r})}{\omega_{j_r} \cdot |s_{i_t} - s_{j_r}|}$$

            **end if**
        **end for**
    **end for**
**end for**
**for** $i \in \{1, 2, ..., |Q|\}$ **do**
    **for** $s_{i_t} \in S_i$ **do**
        **if** $|T_{i_t}| \leq temperature$ **then**
            $s_{i_t} = s_{i_t} + T_{i_t}$
        **else**
            $s_{i_t} = s_{i_t} + \frac{T_{i_t}}{|T_{i_t}|} \cdot temperature$
        **end if**
        **if** *there is a border frame and* $s_{i_t}$ *is outside the frame* **then**
            Bring $s_{i_t}$ on the frame.
        **end if**
    **end for**
**end for**

**Procedure 2:** Summarizes how attractive forces are computed and applied to projected points on each iteration.

The neighbours of the first point of duplication that are in one of the half planes resulted from cutting the plane with a line perpendicular to the axis that produced the maximum pressure and incident on the first point of duplication, are now considered as the neighbours of the second point of duplication and are removed from the neighbours of first point of duplication. The location of the first point of duplication remains unchanged, but the location of the second point of duplication is moved to the centroid of its neighbours. After that any neighbour of the first point of duplication which is closer to the second point of duplication is also considered as the neighbour of the second point of duplication instead.

Let's assume that the point to be duplicated has $c_1$ neighbours before duplication and the first point of duplication has $c_2$ neighbours after duplication and the second point of duplication has $c_3$ neighbours after duplication. Then the value of masses is changed after duplication as follows. The value of the mass for the first point of duplication for applying attractive forces is multiplied by $\frac{c_2}{c_1}$ and the value of the mass for the second point of duplication for applying attractive forces is multiplied by $\frac{c_3}{c_1}$ compared to before duplication. This means that if the point to be duplicated is $S_{i_1}$ and $\omega_{i_1} = \widehat{\omega}$ before duplication, then after duplication:

$$\omega_{i_1} = \frac{c_2}{c_1} \cdot \widehat{\omega} \qquad \text{and} \qquad \omega_{i_2} = \frac{c_3}{c_1} \cdot \widehat{\omega}$$

If modifying the neighbourhood graph results in no neighbours either for the first point of duplication or the second point of duplication, the duplication is considered as a failed duplication. The neighbourhood graph is not changed by a failed duplication as if no attempt was made to duplicate the point. A failed duplication is very unlikely but theoretically it is possible.

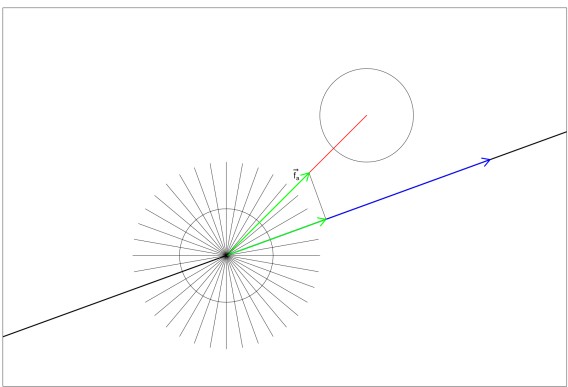

Figure 7: The 36 radial axes around a vertex (point) in G. Perpendicular projection of $\vec{f_a}$ on the blue axis. (Perpendicular projection of $\vec{f_r}$ is similar)

### 4.3.4    The four phases

After the three preliminary steps, the LVSDE algorithm starts the modified force-directed graph drawing through four phases spanning over a number of iterations. By default 1830 iterations are used which is the aggregate sum of the number of iterations of the four phases but small adjustments to the number of iterations should not significantly change the outcome. The algorithm starts with a random embedding with exactly one projection per data instance in which all projected points are in the red layer and no frozen projected point exist[3].

While the first phase is meant to build the general structure of the embedding, the second phase tries to identify the projected points to be moved to the gray layer, freezes them and marks them as ineffective[4]. Phase 3 marks the projected point in the gray layer as effective and reverses the freezing where reversing freezing means the projected points in the red layer become frozen and the projected points in the gray layer are unfrozen[5]. The fourth phase is were the effect of duplication happens on some or all of the projected points in the gray layer. As an example, fig. 8 shows an outcome of the phases of LVSDE on the 1000 genomes project data set Auton et al. (2015) distances at the end of each phase of LVSDE.

### 4.3.5    Phase one: simple graph drawing

Phase one is where exactly one projection per data instance is used and the process runs for a number of iterations (default 500). While in the Frutcherman and Reingold algorithm the temperature changes linearly from maximum temperature ($\bar{u}$) to zero, in the first phase, the temperature is reduced linearly from maximum temperature to half of the maximum temperature. By default temperature in phase one is $\frac{(1000-\mu)\bar{u}}{1000}$ where $\mu$ is the iteration number in current phase. The first iteration starts with a random location of projected points in the visual space and all in the red layer, with none of them frozen or ineffective, and ends in a drawing

---

[3]A frozen projected point means a projected point designated to have no movement and no effect on the movement of other projected points unless the designation is removed.

[4]Once a projected point is designated as ineffective not only does it not move but also it does not apply force on any other projected point. The term freezing a projected point means the projected point does not move but does not necessarily mean that it does not apply forces to other projected points.

[5]An ineffective projected point is a frozen projected point but a frozen projected point is not necessarily a an ineffective projected point.

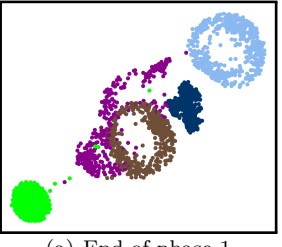 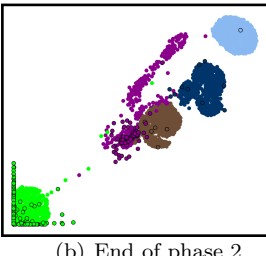 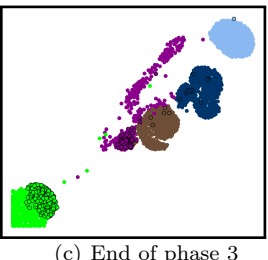 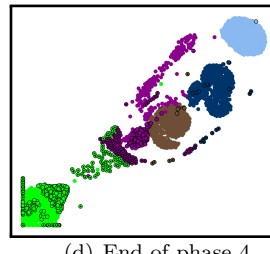

(a) End of phase 1   (b) End of phase 2   (c) End of phase 3   (d) End of phase 4

Figure 8: An LVSDE embedding of the 1000 genomes project data set Auton et al. (2015) distances at the end of each phase of LVSDE. Points with a black circle around them are in the gray layer. Points with a black dot inside them are a second projection of an input data instance. Points without a black circle around them are in the red layer.

which ideally should have balanced forces. The next three phases try to improve the drawing resulted in the phase 1.

In the first phase a border frame is not used in visual space, in order to allow the general structure of the embedding to be shaped in phase one. But at the very beginning of the second phase a frame slightly bigger than the current embedding is put around the embedding which will stay until the end of the algorithm. This is done so that the extra freedom given in the phases 2 to 4 do not enlarge the size of the embedding so high that it requires the embedding to be scaled significantly affecting the visual perception of the embedding negatively.

### 4.3.6   Phase two: selecting gray points and freezing them

The second phase starts with a frame slightly bigger than the current embedding. In the beginning of each iteration of the second phase, a replication pressure is computed for each projected point as was discussed in Section 4.3.3. At the first iteration of the second phase, after replication pressures for the first iteration are computed, the maximum number of points in the gray layer is chosen.

While a good measure for computing replication pressure is achieved in this paper, the discussion on the ideal maximum number of data instances in the gray layer is a more open question. We compute the maximum number of data instances in gray layer as the minimum of two thresholds.

The first threshold is calculated by counting the number of projected points whose replication pressure is outside the range. This range is centred at the mean of replication pressures of all projected points and expanded in each direction 1.2 times the standard deviation of replication pressure of all projected points. The projected points are considered from the embedding at the very beginning of the second phase. While the bidirectional style of this threshold might seem unusual, it is meant to address the cases where the distribution of the replication pressures is twisted toward one direction in a non-symmetric way from the mean.

The second threshold which is the most important one is the number of data instances divided by 4, in order to ensure that the number of data instances in the red layer is not less than 75% of the total number of data instances embedded. In the second phase at each iteration a projected point with the highest replication pressure is chosen and marked as ineffective until as many as the maximum number of data instances in the gray layer are marked as ineffective.

In the second phase when a projected point is marked as ineffective, it is also moved to the gray layer. A projected point in the gray layer remains there until the end of the algorithm but whether it is marked as ineffective or not can change in the next phases.

The second phase continues for a number of iterations (default $450$)[6]. While in the Frutcherman and Reingold algorithm the temperature changes linearly from maximum temperature ($\bar{u}$) to zero, in the second phase, the temperature is reduced linearly from half of the maximum temperature to almost zero[7]. By default the temperature in phase two is $\frac{(1000-(\mu+500))\bar{u}}{1000}$ where $\mu$ is the iteration number in current phase.

### 4.3.7 Phase 3: reversing freezing and marking gray points as effective

At the very beginning of the third phase, every projected point that was marked as ineffective in the second phase is marked as effective again but the rest of the projected points are frozen. This also implies reversing the freezing as the projected points in the red layer become frozen but effective and the projected points in the gray layer are unfrozen. The third phase continues for a number of iterations (default 390). While in the Frutcherman and Reingold algorithm the temperature changes linearly from maximum temperature ($\bar{u}$) to zero, in the third phase, the temperature is reduced linearly from almost half of the maximum temperature to almost zero. By default the temperature in phase three is $\frac{(1000-(\mu+510))\bar{u}}{1000}$ where $\mu$ is the iteration number in current phase.

### 4.3.8 Phase 4: replicating gray points when possible

In the first iteration of phase 4, an attempt is made to duplicate each of the projected points of the gray layer. When duplicating a projected point, the new added point is also set to be in the gray layer. When duplicating, the neighbourhood graph is also modified as was discussed in Section 4.3.3. Not all duplication are successful and a failed duplication may happen when trying to modify the neighbourhood graph. A failed duplication does not change the embedding or neighbourhood graph. In the proposed algorithm a maximum of 2 projections per data instance limit is enforced.

The force-directed graph drawing continues in the forth phase through a number of iterations (default 490). While in the Frutcherman and Reingold algorithm the temperature changes linearly from maximum temperature ($\bar{u}$) to zero, in the fourth phase, the temperature is reduced linearly from almost half of the maximum temperature to zero. By default the temperature in phase four is $\frac{(1000-(\mu+510))\bar{u}}{1000}$ where $\mu$ is the iteration number in current phase.

### 4.4 Parallel speedup

A parallel version of the proposed algorithm is possible and is implemented[8] which scales well with an increase in the number of CPU cores. In order to parallelized the algorithm, the For Loops for attractive and repulsive forces should be parallelized. If coded by slicing the calculations correctly, a race condition in the calculation of the forces can be avoided.

### 4.5 Parameters

The visual density adjustment parameter ($b$), the number of neighbours for building neighbourhood graph ($\hat{p}$), and whether to use UMAP to 30 dimensions as a preliminary step can significantly change the outcome of the algorithm.

To overcome the challenge of setting the values of parameters, one approach that provides practical benefit is as follows. While the visual density adjustment parameter can be set freely to any real number and while the number of neighbours for building neighbourhood graph can be set freely to any positive integer, in practice constraints can be used. Choosing the visual density adjustment parameters from the set $\{-0.9, -0.5, -0.1, 0.1, 0.5, 0.9\}$ and the number of neighbours for building neighbourhood graphs from the set $\{10, 20, \lfloor\frac{|Q|}{3}\rfloor, \lfloor\frac{|Q|}{4}\rfloor, \lfloor\frac{|Q|}{5}\rfloor\}$ where $|Q|$ is the number of data instances in original space empirically provide a practical way to approach the challenge.

---

[6]Using 450, 390 & 490 as the default values in the phases two, three and four, remained for historical reasons in the development of the algorithm but other values of a similar scale such as 500 would also work as well.

[7]Almost zero and almost half for some of temperature limits in phases two, three and four were used in the code base for historical reasons as well, these can also be replaced with simply zero and half.

[8]https://web.cs.dal.ca/~barahimi/chocolate-lvsde/

In this way, by testing the outcome of a total of 60 configurations for the three parameters sufficiently useful values for the parameters may be found. For text related high dimensional data sets, whether to use Euclidean distance or Cosine distance is a consideration that needs to be added to the above discussion as Cosine distance in general is known to usually perform better on text based high dimensional data.

## 5 Experimental Setup

### 5.1 Data Sets

In this paper six different data sets are considered. The first data set is the 1000 genomes project Auton et al. (2015) consisting of DNA samples from 2504 individuals in different locations in the world. The data from the third phase of the project is used and converted to a distance matrix (details in subsection 5.2). The second data set is the MNIST set of handwritten digits (details in subsection 5.2). The third data set is the IRIS data set classifying 3 different classes of the Iris plant. The fourth is the MeefTCD Barahimi (2022) enhanced email features data set and has 2400 points and 48557 dimensions and is labelled with 8 classes described in Barahimi (2022). The fifth and sixth are neural network layer outputs on the CIFAR-10 Krizhevsky image classification data set.

### 5.1.1 CIFAR-10 Image Classifier Neural Network Last Hidden Layer

For the fifth and sixth data sets, instead of using dimensionality reduction on the CIFAR-10 Krizhevsky data set directly, we first trained a neural network image classifier using the data. Then for the fifth data set we took the output of the last hidden layer on the first 5000 data instances and for the sixth data set we took the output of the last hidden layer on the first 5000 data instances of the test data.

The architecture is a Convolutional Neural Network (CNN) displayed in Fig. 9. It consists of two convolutional layers each with $2 \times 2$ maxpooling and ReLU activation, one with 32 filters and the second one with 16 filters, then a Dropout with probability of 0.5, a flattening layer and then two fully connected 800 dimensional DropConnect Wan et al. (2013) layers. Each has a DropConnect probability 0.5 and ReLU activation. Finally there is a fully connected 10 dimensional layer with softmax activation. It is notable that the architecture, while reasonable, is not the state of the art in image classification as the purpose of this paper is evaluation of a dimensionality reduction method.

The training is done in 100 epochs using the Adam Kingma & Ba (2014) optimizer of TensorFlow and all training data of CIFAR-10.

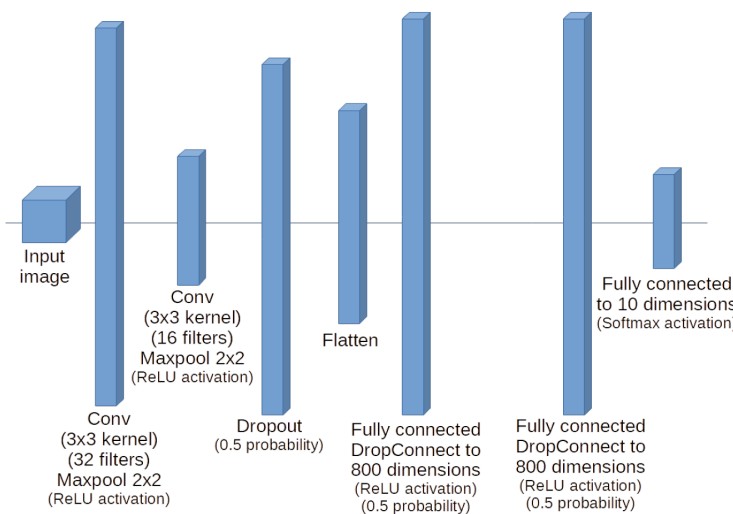

Figure 9: Convolutional Neural Network (CNN) image classifier architecture for CIFAR-10

### 5.2 Experimental Setup Details

The proposed method (*LVSDE*) is evaluated in this paper in comparison to UMAP McInnes et al. (2018a;b) and the t-SNE Maaten & Hinton (2008) Barnes-Hut variant Van Der Maaten (2014) on the six data sets. For UMAP McInnes et al. (2018a;b), the umap-learn 0.5.3 python implementation of UMAP McInnes et al. (2018a;b) is used.

For the 1000 genomes project data set Auton et al. (2015), for the *LVSDE* method, two different configurations are reported on and the difference between them is discussed. For the first configuration, the parameters that are used are the visual density adjustment parameter $b = 0.1$, one third of the number of data instances as the number of neighbours in the initial neighbourhood graph, and opting not to use UMAP to 30 dimensions. For the second configuration, the parameters that are used are visual density adjustment parameter $b = 0.5$, 20 as the number of neighbours in the initial neighbourhood graph, and opting not to use UMAP to 30 dimensions.

The data from the third phase of the 1000 genomes project is used and converted to a distance matrix using the version 2.0 of the PLINK Purcell et al. (2007); Purcell whole genome analysis toolkit. This has an implementation of the Yang et al. (2011) method for producing a similarity relationship matrix followed by a simple linear transformation to produce non-negative dissimilarity distances (maximum similarity minus the similarity as distance), and the distance matrix is used as the input to all methods.

For the MNIST data set, for the *LVSDE* method the parameters that are used are visual density adjustment parameter $b = 0.1$, and 50 as the number of neighbours in the initial neighbourhood graph, opting to use UMAP to 30 dimensions. Also for LVSDE, Euclidean distance is used for UMAP to 30 dimensions and is used to convert 30 dimensional data of UMAP output to distances.

For the MNIST data set, for *LVSDE*, all MNIST training data are embedded into 30 dimensions but only the first 5000 rows are used for the rest of the algorithm, evaluation and display. Similar options are chosen for UMAP where UMAP is applied to all training data but only the first 5000 rows are picked for evaluation and display. Similar conditions are used for t-SNE where t-SNE is applied to all of training data but then only the first 5000 rows are picked for evaluation and display. For the MNIST data set, for all methods, Euclidean distance is used as the distance measure.

For the IRIS data set the parameters that are used for *LVSDE* are the visual density adjustment parameter $b = -0.1$, and 20 as the number of neighbours in the initial neighbourhood graph, and opting not to use UMAP to 30 dimensions. For all methods, Euclidean distance is used as the distance measure.

For the MeefTCD email features data set, for the *LVSDE* method, the parameters that are used were the visual density adjustment parameter $b = 0.5$, 40 as the number of neighbours in the initial neighbourhood graph, opting to use UMAP to 30 dimensions. Also for LVSDE, Cosine distance is used as the distance measure for UMAP to 30 dimensions but afterthat, Euclidean distance is used. For t-SNE and UMAP Cosine distance is used as the distance measure.

For the neural network output on the CIFAR-10 data set, for the *LVSDE* method, two different configurations are reported on and the difference between them is visualized. For the first configuration, the parameters that are used are visual density adjustment parameter $b = 0.1$, and 20 as the number of neighbours in the initial neighbourhood graph, opting to use UMAP to 30 dimensions. For the second configuration, the parameters that are used are visual density adjustment parameter $b = 0.5$, and 20 as the number of neighbours in the initial neighbourhood graph, opting to use UMAP to 30 dimensions. Euclidean distance is used for for all configurations.

## 6   Visual metaphors

To visualize a Strict Red Gray Embedding, several visual metaphors are used in this paper. For example, in the simplest form, the points in the red layer are rendered with red colour and the points in gray layer are rendered with gray colour. Another example is that giving priority to one layer when overlap occurs, meaning that first the points of one of the layers are rendered and then the points of the other layer are

rendered on top of them. To be able display class colours in addition to the separation of layers, a visual metaphor is used where class colours are used as the rendering colour for the points of visual space but a black circle is drawn around points of only one of the layers, or points of only one of the layers are drawn smaller. Fig. 13(a,b) is a good showcase of these metaphors.

Another variant that is considered for grayscale images especially for the readability of MNIST digits is to draw images for the points in the red layer by a linear transformation of grayscale values $[0 - 255]$ to the range of colours starting with full black and ending to full red. For the points in the gray layer however, the linear transformation that transforms $[0 - 255]$ grayscale values to the range of colours starting from mid gray colour to full white is considered. See Fig. 2 as a showcase.

## 6.1 Visually connecting multiple projections of data instances

We wish to visually connect multiple projections of a data instance for all data instances that have more than one projection. In the simplest form, where there are at most two projections per data instance, the connection can be a line between the two projections of the data instance. Conceptually this can be modelled by a graph where an edge between two projected points indicate that they are projections of the same data instance. In this scenario, if the visual metaphor can visually connect on a group basis rather than individual data instances, then less clutter is caused in the visualization. For example, if the number of data instances that have two projections is 300, and if the viewer has to think about them individually, there are 300 pairs of projected points which takes a significant amount of time even if somehow those 300 connections between the 300 pairs of projected points were visualized without clutter.

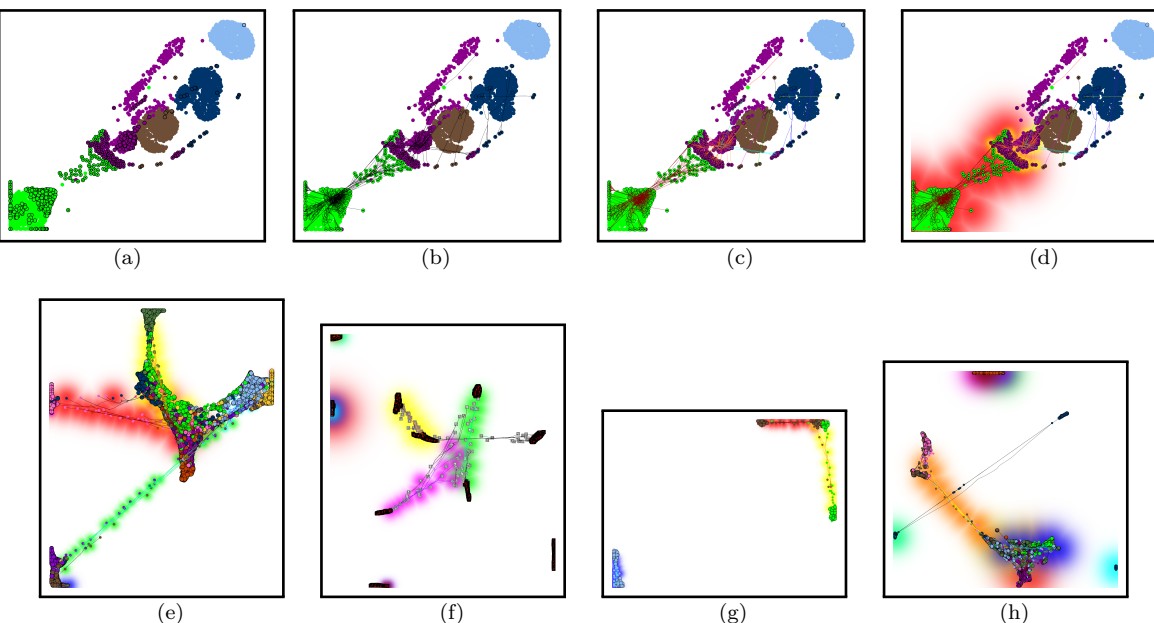

Figure 10: (a,b,c,d) An LVSDE embedding of the 1000 genomes project data set Auton et al. (2015) distances with different visual metaphors.(b) Edge bundling algorithm of Holten & Van Wijk (2009) is applied on the duplication graph. (c) Edge clustering is done based on the edge bundling. (d,e,f,g,h) Meta balls are used to visualize edge clusters which are based on the edge bundling done by the algorithm of Holten & Van Wijk (2009). (a,b,c,d) Points with a black circle around them are in gray layer. Points with a black dot inside them are second projection of an input data instance. Points without a black circle around them are in red layer. (e,g,h) Points with a black circle around them are in the red layer and the points in the gray layer are drawn smaller. (f) An LVSDE embedding of a subset of the MNIST data set visualized with meta balls. (g) An LVSDE embedding of a subset of the IRIS data set visualized with meta balls. (h) An LVSDE embedding of a subset of the MeeefTCD data set visualized with meta balls.

To reduce the clutter and provide a means of grouping for the pairs of projected points the following approach is proposed. The approach starts with the edge bundling algorithm of Holten & Van Wijk (2009) which draws each edge of the graph as a multi-segment polyline in a way that related edges overlap on each other which reduces clutter. Next, edge clustering partitions the set of the drawn edges into multiple sets called edge clusters setting the stage for grouping pairs of projected points (see Fig. 10(c) for an example).

Although edge bundling reduces clutter, there is still room for improvement. Next we use an adaptation of the concept of meta balls Blinn (1982) to guide the viewer on which parts of the edge bundling and edge clustering to examine, and secondly provide an approximate partition of visual space (see Fig. 10(d) for an example). Meta balls, are algebraic surface drawings Blinn (1982) meaning that an algebraic formula is used to specify the colour of each pixel of a surface. In the adaptation that we use in this paper however, a meta ball has a corresponding edge cluster and instead of having one center, a single meta ball has multiple centers which are the endpoints of the edges in the cluster. The density change of the meta ball is reflected in the opacity component of the RGBA colour of each pixel and the size of the edge cluster also affects how much the meta ball spreads in the visual space. This allows the signifying of larger clusters.

The meta balls are only displayed for the ten largest size edge clusters and a different base colour is assigned to each meta ball. Iterating through all meta balls in the order of corresponding edge cluster size, for each meta ball, the pixels of the meta ball are drawn on a separate canvas for each meta ball. The canvases are then displayed on top of each other with the opacity of pixels controlling how much the previous canvases' pixels should be visible. Then the actual projected points are drawn and then the corresponding edge bundling are drawn.

Consider again Fig. 10, in which edge bundling, edge clustering and adapted meta balls are used to visualize embeddings of different data sets. In this example we want to emphasize the blue meta ball in Fig. 10(f) which guides the user to the part of visual space where duplication of the digit 8 has occurred. Next the edge bundling in that region fine tunes the picture of the path on which duplications of digit 8 are located.

With regards to the details, when applying the edge bundling algorithm of Holten & Van Wijk (2009), we avoid the smoothing of edges discussed in Holten & Van Wijk (2009). Moreover, in performing the edge clustering, approximate edge distances and the concept of edge compatibility defined in Holten & Van Wijk (2009) are used as a measure to combine edge clusters. Initially, each edge is a separate cluster and if two edges belonging to two different clusters are found that have both a sufficiently small approximate distance and sufficiently large edge compatibility as defined in defined in Holten & Van Wijk (2009), then the two clusters are combined to one cluster. Furthermore, when choosing the ten largest clusters, if the number of edge clusters is less than ten, every edge cluster is selected. And most notably, if the RGB colour assigned to a cluster is specified by $(R_1, G_1, B_1)$ and if the coordinates of endpoints of the edges of the cluster are specified by $\{(x_1, y_1), (x_2, y_2), ..., (x_\xi, y_\xi)\}$ then the RGBA colour of a pixel of the adapted meta ball at the coordinates $(x, y)$ is defined as

$$\left( R_1, G_1, B_1, \frac{0.8}{EXP\left( \frac{\min\limits_{1 \leq i \leq \xi} \left( (x-x_i)^2 + (y-y_i)^2 \right)}{\left( \beta \cdot \frac{\xi}{\xi_{max}} \right)^2} \right)} \right)$$

where $\beta$ indicates the spread of the metaball for the largest edge cluster and $\xi_{max}$ specifies the size of the largest cluster. Since the opacity as it is specified above is at most 80%, overlapping meta balls can still be perceivably detectable to some degree.

## 7 Empirical Results And Comparison

### 7.1 Qualitative Analysis of Results

Starting with qualitative results, the most notable result in terms of semantics is the embedding of the 1000 genomes project data set Auton et al. (2015) distances by LVSDE displayed in Fig. 1. The 1000 genomes

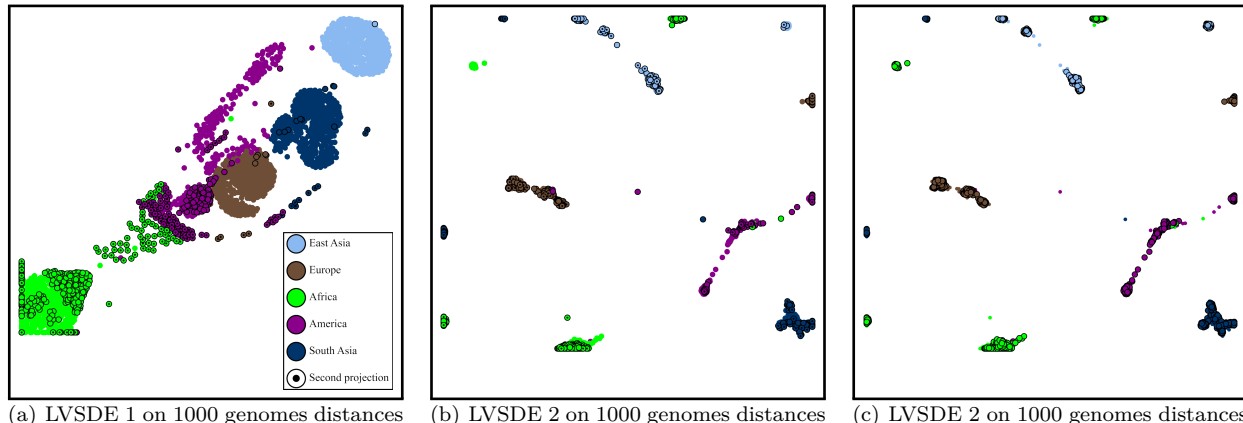

(a) LVSDE 1 on 1000 genomes distances     (b) LVSDE 2 on 1000 genomes distances     (c) LVSDE 2 on 1000 genomes distances

Figure 11: Two configurations of LVSDE used for embedding the 1000 genomes project data set Auton et al. (2015) distances. While configuration 1 performs better semantically, configuration 2, like UMAP and t-SNE artificially performs better in groups separation, however knowing the phylogenetic connection of human genomes that separation has less semantic harmony with the reality of significant connection between human genomes. Configuration 1 although being better does not show the ideal representation of connection human genomes either and for future work it is an interesting research area to improve that in order to have an embedding method that shows the connection of human genomes better and more inclusive. The shortcoming in representation of that connection is an important flaw in the current state dimensionality reduction methods improved by LVSDE but not to the full extent. In part a and b, points with a black circle around them are in gray layer and points with a black dot inside them are second projection of an input data instance and points without a black circle around them are in red layer. In part c, points with a black circle around them are in the red layer and the points in the gray layer are drawn smaller.

project is one of the first genomes data sets for studying human genome variety and contains the DNA samples from different locations of the world labelled with geographical locations.

One important observation in Fig. 1 is that some of the projected points labelled with America are duplicated from some area close to the area of visual space consisting of the projected points labelled with Europe to an area close to duplicates of some of projected points labelled with Africa. This matches the immigration history from Europe and Africa to America. In this manner LVSDE has achieved multiple successes. One is identifying points to duplicate and second, duplicating them in a meaningful way to a correct area of visual space showing its capacity to guide the duplication. While Figs. 12(a) and 12(b) show embeddings of 1000 genomes projected distances using UMAP and t-SNE, the fact that they do not support duplication, puts them in a weaker position in terms of meaning.

In Fig. 11 two configurations of LVSDE are used for embedding the 1000 genomes project data set Auton et al. (2015) distances. While configuration 1 performs better semantically, configuration 2, like UMAP and t-SNE in Figs. 12(a) and 12(b), unrealistically separates the genomes in a way that hides the significant connection between human genomes regardless of location or race. Knowing the phylogenetic connection of human genomes regardless of any division, that separation has less semantic harmony with the reality. Configuration 1 although being better does not show the ideal representation of connection between human genomes either and for future work it is an interesting research area to improve that in order to have an embedding method that shows the connection more clearly and inclusively.

The next important result is LVSDE on a subset of the MNIST data set. For the MNIST data set, it is known that popular dimensionality reduction methods have some difficulty in separating digits 3, 5 and 8. The handwritten digits do not necessarily correspond to digital digits with complete certainty. A handwritten digit, especially if not written well may be for example 80% 3, 60% 5 and 30% 8, the so called multivariate fuzzy semantics that was discussed in Section 2. But digits 3, 5 and 8 are not the only digits that cause issues for dimensionality reduction techniques on the MNIST data set.

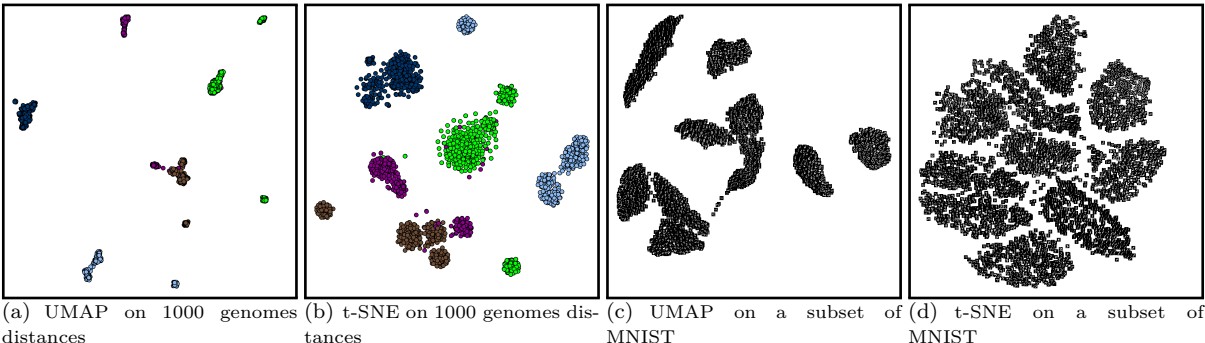

(a) UMAP on 1000 genomes distances

(b) t-SNE on 1000 genomes distances

(c) UMAP on a subset of MNIST

(d) t-SNE on a subset of MNIST

Figure 12: UMAP and t-SNE embeddings of 1000 genomes project data set Auton et al. (2015) distances and a subset of the MNIST data set.

Digits 4 and 9 also confuse dimensionality reduction techniques limiting the ability of such techniques to separate them. Again, the case of multivariate fuzzy semantics exists for some of the handwritten digits of 4 and 9. Looking at Fig. 2, the separation of digits 3, 5 and 8 on the red layer and also the separation of digits 4 and 9 on the red layer are better than popular techniques. By using layers, the identification of groups in the LVSDE embedding of the subset of MNIST data set is much easier when only considering the red layer while the gray layer provides some background information. For comparison, Figs. 12(c) and 12(d) show UMAP and t-SNE embeddings of the subset of the MNIST data set. While t-SNE does a better job than UMAP on separating digits 3, 5 and 8, it has a lot of misplaced digits 3, 5 and 8 meaning a digit being in close proximity to images of different digits.

Next in line is the IRIS data set where detecting subgroups is the focus. Identifying a subgroup means identifying a significantly large subset of a group that only contains data instances from that group but not all of them. Being a very simple data set, IRIS has only three classes for groups. By looking at Fig. 13(b) which does not have class colours, one would normally immediately identify at least 4 subgroups, out of which only one is wrong, meaning at most a 25% chance of failure. In contrast, if class colours are removed from Fig. 13(c) for t-SNE on IRIS, one would normally identify only two subgroups, out of which one of them is wrong, meaning a 50% chance of failure. While UMAP on IRIS shown in Fig. 13(d), is not as defected as t-SNE, how it compares to LVSDE can be a subjective argument and requires more analysis.

For the MeeefTCD data set, identification of combinational groups is the focus. In Fig. 14, the separation of the combination of two classes CES and logistics is much better for LVSDE than t-SNE and UMAP since for LVSDE, the two classes combined are clearly separate from the rest of classes but for t-SNE the two classes combined are not separate from at least 5 other classes and for UMAP, the two classes combined are not separate from 4 other classes.

For the image classifier neural network on CIFAR-10, we took the output of last hidden layer (the second DropConnect layer) for first 5000 data instances of the training data and passed it to LVSDE, t-SNE and UMAP which yielded the embeddings displayed in Figs. 15 and 16. We also took the output of last hidden layer (the second DropConnect layer) for the first 5000 data instances of the test data and passed it to LVSDE, t-SNE and UMAP which yielded the embeddings displayed in Figs. 17 and 18. Looking at Figs. 15 and 16, better separation of some of the classes by LVSDE when compared to t-SNE and UMAP is very clear. Looking at Figs. 17 and 18, at last for configuration 1 of LVSDE, the argument holds, especially if someone combines some of the classes together.

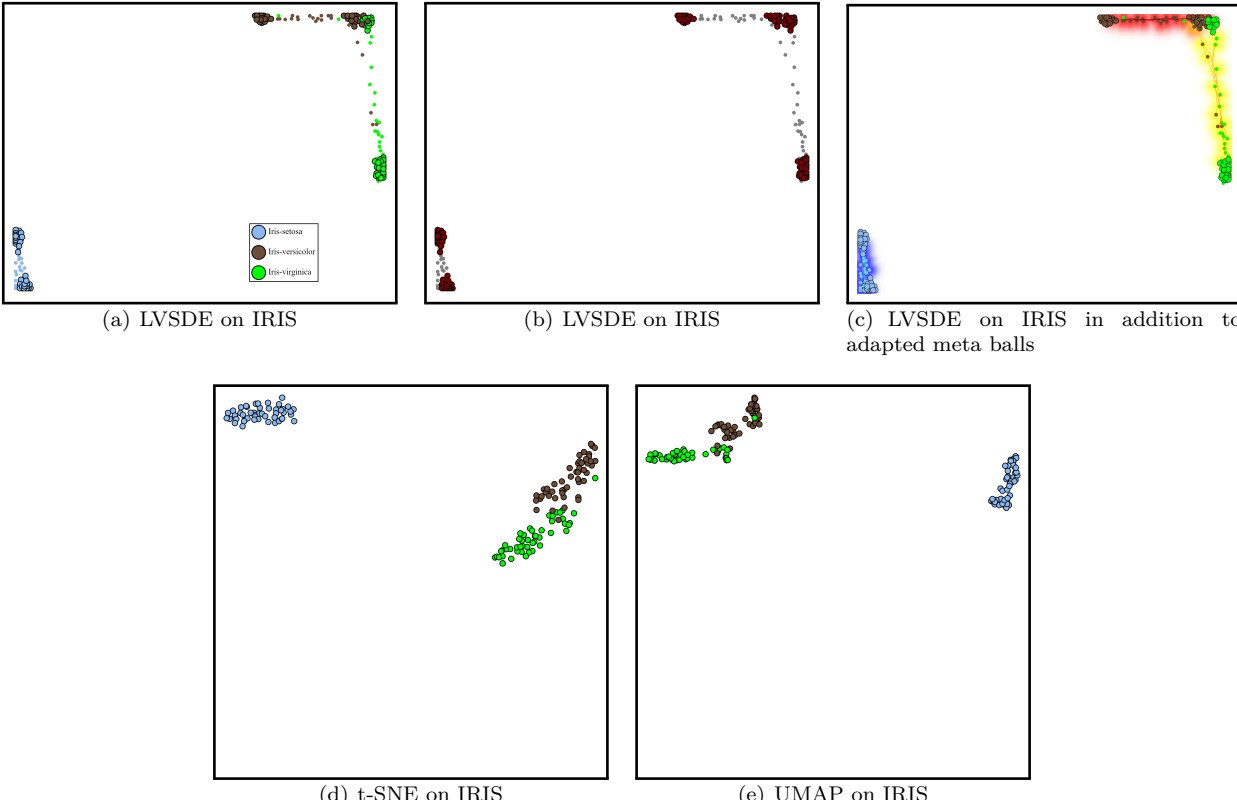

(a) LVSDE on IRIS       (b) LVSDE on IRIS       (c) LVSDE on IRIS in addition to adapted meta balls

(d) t-SNE on IRIS       (e) UMAP on IRIS

Figure 13: Embeddings of the IRIS data set using different dimensionality reduction methods. For LVSDE points with a black circle around them are in the red layer and the points in the gray layer are drawn smaller. For LVSDE in part b, points in the red layer are coloured red and the points in the gray layer are coloured gray. LVSDE has performed better for subgroup detection.

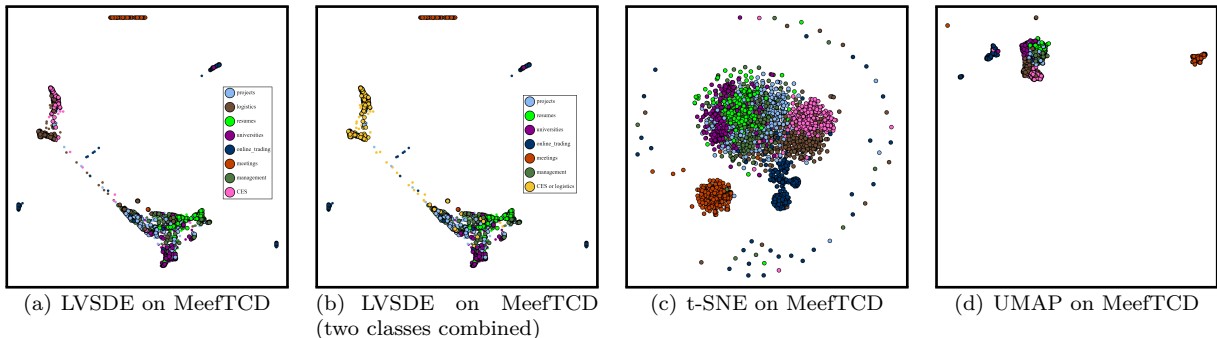

(a) LVSDE on MeefTCD    (b) LVSDE on MeefTCD (two classes combined)    (c) t-SNE on MeefTCD    (d) UMAP on MeefTCD

Figure 14: Embeddings of the MeefTCD data set using different dimensionality reduction methods. For LVSDE points with a black circle around them are in the red layer and the points in the gray layer are drawn smaller. For LVSDE in part b, two of the classes are combined. The separation of the combination of two classes CES and logistics is much better in the LVSDE embedding.

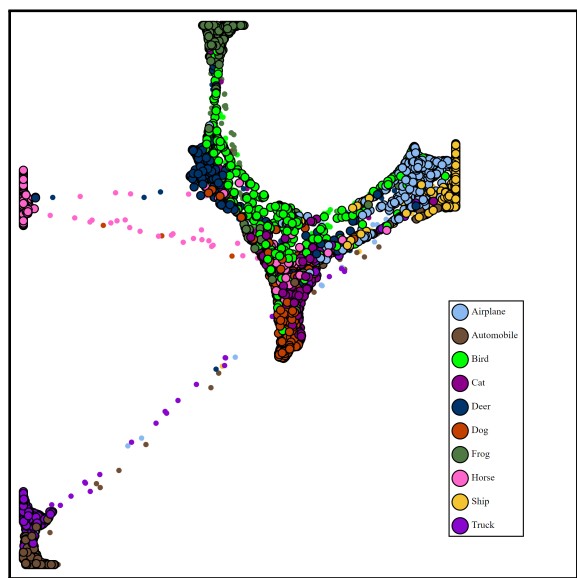

(a) LVSDE configuration 1 on a subset of image classifier neural network layer output on CIFAR-10 training data.

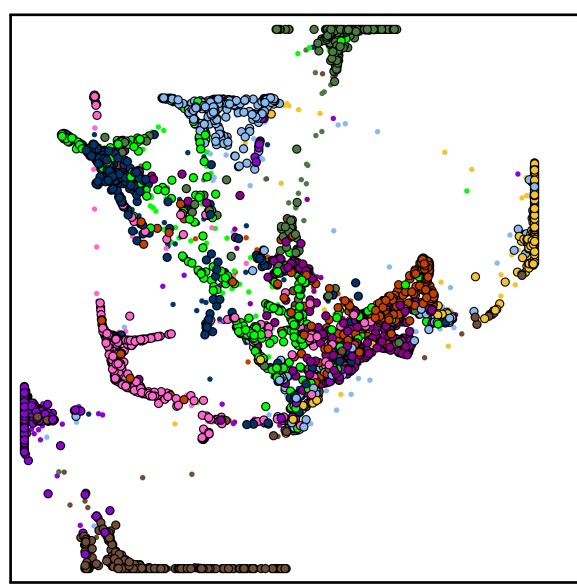

(b) LVSDE configuration 2 on a subset of image classifier neural network layer output on CIFAR-10 training data.

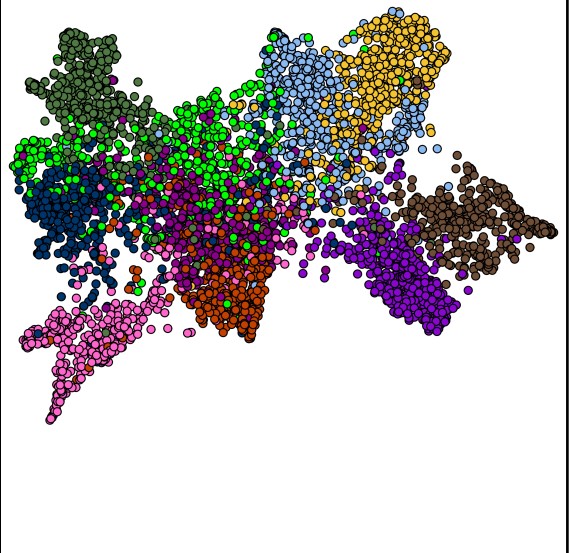

(c) t-SNE on a subset of image classifier neural network layer output on CIFAR-10 training data.

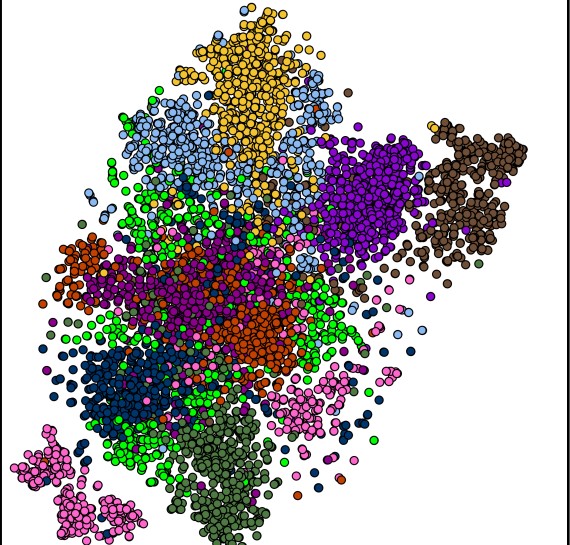

(d) UMAP on a subset of image classifier neural network layer output on CIFAR-10 training data.

Figure 15: Embeddings of a subset of image classifier neural network layer output on CIFAR-10 training data, using different dimensionality reduction methods. For LVSDE points with a black circle around them are in the red layer and the points in the gray layer are drawn smaller.

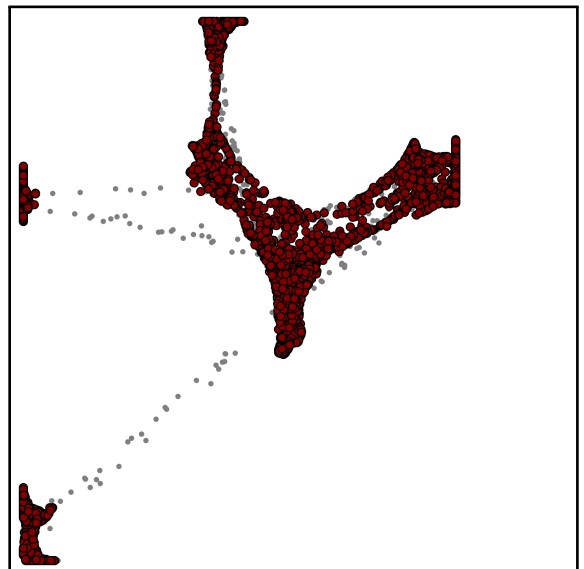

(a) LVSDE configuration 1 on a subset of image classifier neural network layer output on CIFAR-10 training data.

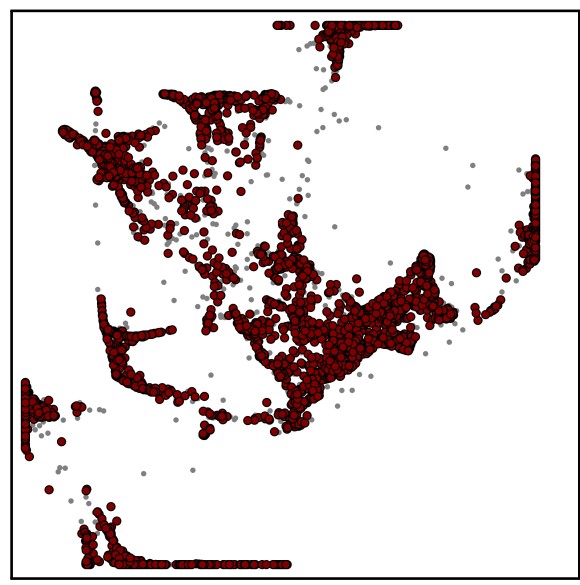

(b) LVSDE configuration 2 on a subset of image classifier neural network layer output on CIFAR-10 training data.

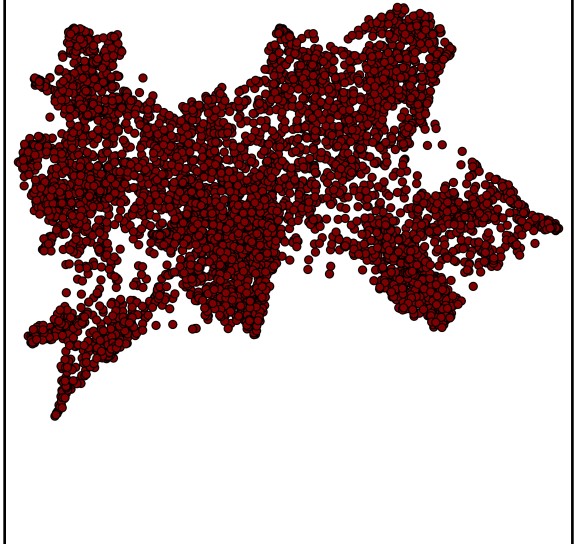

(c) t-SNE on a subset of image classifier neural network layer output on CIFAR-10 training data.

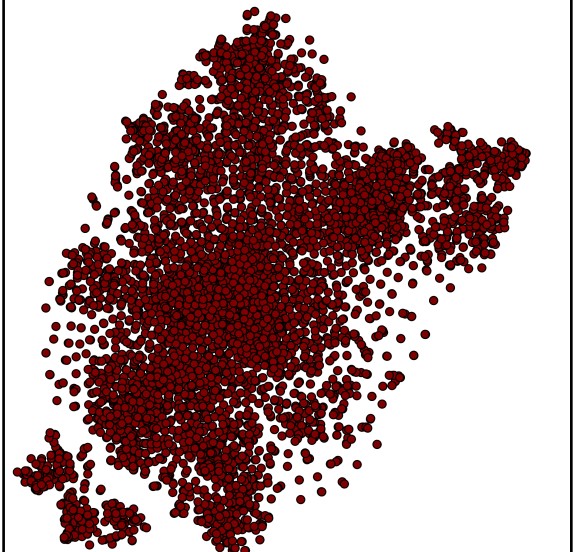

(d) UMAP on a subset of image classifier neural network layer output on CIFAR-10 training data.

Figure 16: Embeddings of a subset of image classifier neural network layer output on CIFAR-10 training data, using different dimensionality reduction methods. For LVSDE points with a black circle around them are in the red layer and the points in the gray layer are drawn smaller. For LVSDE points in the red layer are coloured red and the points in the gray layer are coloured gray.

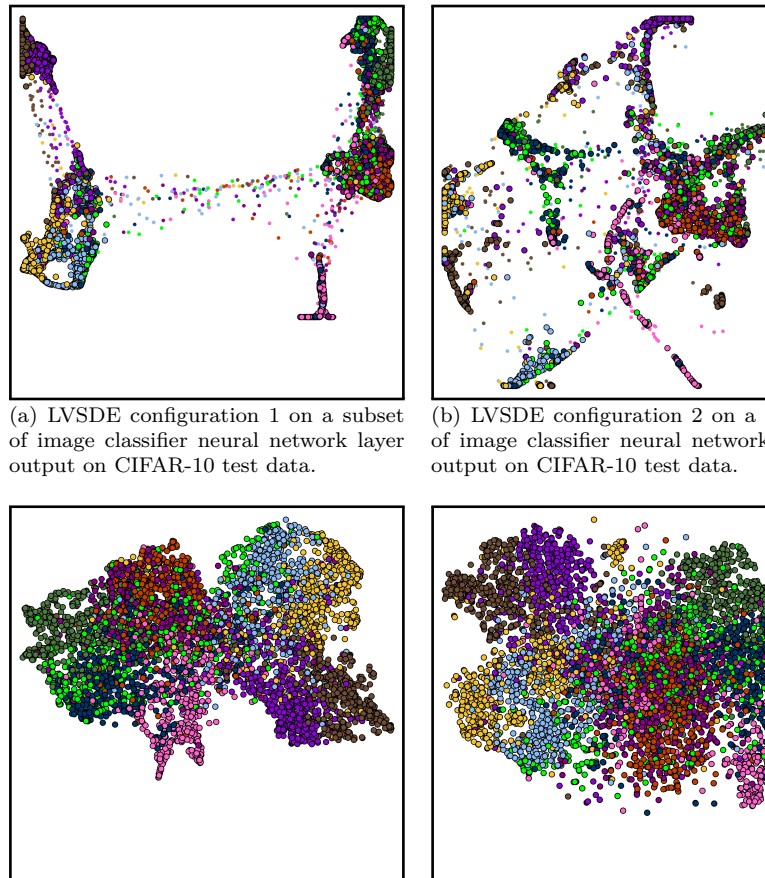

(a) LVSDE configuration 1 on a subset of image classifier neural network layer output on CIFAR-10 test data.

(b) LVSDE configuration 2 on a subset of image classifier neural network layer output on CIFAR-10 test data.

(c) t-SNE on a subset of image classifier neural network layer output on CIFAR-10 test data.

(d) UMAP on a subset of image classifier neural network layer output on CIFAR-10 test data.

Figure 17: Embeddings of a subset of image classifier neural network layer output on CIFAR-10 training data, using different dimensionality reduction methods. For LVSDE points with a black circle around them are in the red layer and the points in the gray layer are drawn smaller.

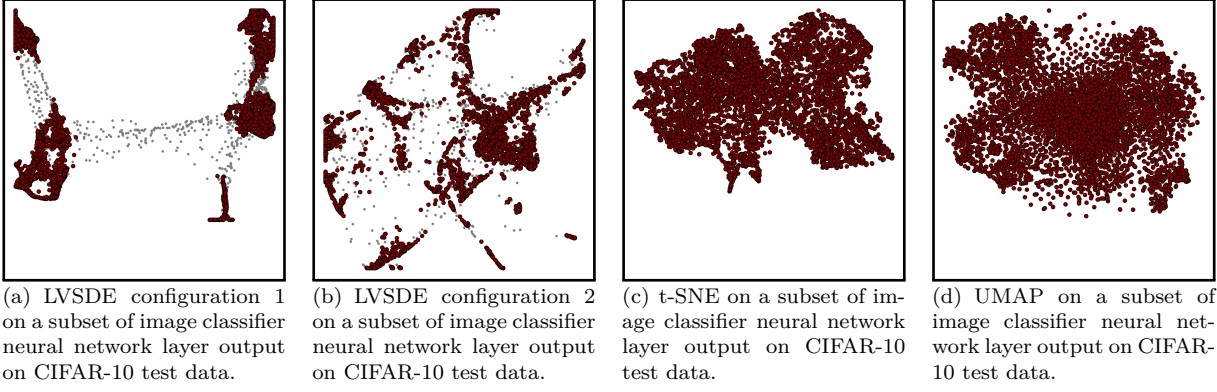

(a) LVSDE configuration 1 on a subset of image classifier neural network layer output on CIFAR-10 test data.

(b) LVSDE configuration 2 on a subset of image classifier neural network layer output on CIFAR-10 test data.

(c) t-SNE on a subset of image classifier neural network layer output on CIFAR-10 test data.

(d) UMAP on a subset of image classifier neural network layer output on CIFAR-10 test data.

Figure 18: Embeddings of a subset of image classifier neural network layer output on CIFAR-10 training data, using different dimensionality reduction methods. For LVSDE points with a black circle around them are in the red layer and the points in the gray layer are drawn smaller. For LVSDE points in the red layer are coloured red and the points in the gray layer are coloured gray.

## 7.2 Quantitative Measurement and Qualitative Analysis

While embeddings can be visually evaluated qualitatively, it is good to also have some quantitative measurement of the embeddings. The objectives of qualitative analysis and quantitative are not necessarily the same as they evaluate different aspects of the embeddings and quantitative analysis alone should not be considered as the sole indicator of better embeddings but can increase the confidence of the reader in the qualitative visual results.

Since Multi-layered Multi-point Dimensionality Reduction and therefore Strict Red Gray Embeddings are not completely compatible with existing quantitative measurements some discussion is needed on how to adapt current quantitative measurements for them. In particular a measure called $\Lambda$ measure is used in this paper which is the same as KNN classification accuracy except adapted to Multi-layered Multi-point Dimensionality Reduction and Strict Red Gray Embeddings.

For a layer $\theta$ (or set of layers $L$) of a Multi-layered Multi-point Dimensionality Reduction embedding, the $\Lambda$ measure is the percentage (or ratio) of data instances that have at least one projection in $\theta$ (or $L$) and the class label for that data instance matches the label with maximum occurrence among $k$ nearest neighbours of any of the projections of that data instance in $\theta$ (or $L$). Neighbours are limited to a specific set of layer(s) in visual space called classification layer(s) $\widehat{L}$ and the result of evaluation is denoted by $\Lambda_{(\theta,\widehat{L})}$ or $\Lambda_{(L,\widehat{L})}$ respectively ($k$ is called evaluation neighbourhood size, $\theta$ or $L$ are called evaluation layer(s). The choice of $k$, $\theta$ or $L$ and $\widehat{L}$ are parameters for evaluation. In this paper $k = 15$ is used the evaluation neighbourhood size for all experiments of the paper).

Table 1 shows the quantitative results and comparison on the six data sets using different dimensionality reduction methods. Three different combinations of evaluation layers and classification layers are reported since. While red-red evaluation is more suitable when the intent is finding a sufficient portion of data that shows a pattern, the all-all evaluation is more suitable when the intent is finding a pattern that covers all of the data. Gray-gray evaluation when compared to red-red evaluation, is more suitable to see how the gray layer and the red layer are acting differently.

| Data set | Embedding method | Measure 1 (red-red) | Measure 2 (all-all) | Measure 3 (gray-gray) |
|---|---|---|---|---|
| 1000 genomes | LVSDE configuration 1 | 98.551% | 98.562% | 97.802% |
| 1000 genomes | LVSDE configuration 2 | 99.708% | 99.681% | 99.333% |
| 1000 genomes | UMAP | 99.321% | 99.321% | NA |
| 1000 genomes | t-SNE | 99.241% | 99.241% | NA |
| MNIST | LVSDE | 97.359% | 97.040% | 90.492% |
| MNIST | UMAP | 97.000% | 97.000% | NA |
| MNIST | t-SNE | 96.400% | 96.400% | NA |
| IRIS | LVSDE | 97.368% | 95.333% | 86.111% |
| IRIS | UMAP | 97.333% | 97.333% | NA |
| IRIS | t-SNE | 97.333% | 97.333% | NA |
| MeeefTCD (classes not combined) | LVSDE | 74.359% | 74.542% | 72.222% |
| MeeefTCD (classes not combined) | UMAP | 75.083% | 75.083% | NA |
| MeeefTCD (classes not combined) | t-SNE | 75.125% | 75.125% | NA |
| Neural network layer on training data | LVSDE configuration 1 | 79.121% | 78.560% | 70.000% |
| Neural network layer on training data | LVSDE configuration 2 | 79.451% | 78.860% | 72.222% |
| Neural network layer on training data | UMAP | 76.580% | 76.580% | NA |
| Neural network layer on training data | t-SNE | 79.040% | 79.040% | NA |
| Neural network layer on test data | LVSDE configuration 1 | 63.165% | 62.100% | 47.778% |
| Neural network layer on test data | LVSDE configuration 2 | 61.692% | 61.600% | 50.889% |
| Neural network layer on test data | UMAP | 60.740% | 60.740% | NA |
| Neural network layer on test data | t-SNE | 61.440% | 61.440% | NA |

Table 1: Experiments' *knn* measures on the six data sets. For the column Measure 1, the evaluation layer is the red layer and the classification layer is the red layer. For the column Measure 2, the evaluation layers are both the red layer and the gray layer while the classification layers for Measure 2 are also both the red layer and gray layer. For the column Measure 3, the evaluation layer is the gray layer and the classification layer is the gray layer. For t-SNE and UMAP it is assumed that all projected points are in red layer. All the numbers in this table are rounded to 3 decimal places. Evaluation neighbourhood size $k = 15$ is used.

## 8  Availability

For an implementation of LVSDE, please visit the URL `https://web.cs.dal.ca/~barahimi/chocolate-lvsde/` (date accessed: January 12, 2023).

## 9  Limitations

One challenging and not completely resolved potential of Multi-layered Multi-point Dimensionality Reduction is the preservation of a well-defined topology structure where the meaning is interpreted in the context of finite point set topology Evans et al. (1967); Aleksandrov (1998). While infinite geometric topology Sher & Daverman (2001) or infinite shape theory Kendall et al. (2009) are also conceptually relevant as a context for topology, their relevance is less direct than finite point set topology which deals with a finite number of points. If the challenge of visually connecting multiple projections of a point is adequately addressed, preservation of well defined topology structures can become a significant potential for Multi-layered Multi-point Dimensionality Reduction. The preservation is in a sense that a well defined procedure finds a topology structure in original space and a corresponding well defined visual procedure finds a close or exactly the same topology structure in visual space.

Furthermore, LVSDE is dependent on some parameters which can significantly change the outcome of the embedding. While the diversity of possible embeddings is actually in favour of LVSDE, lack or weakness of existence of an established way to set those parameters is a limitation.

For the 1000 genomes project data set, on the inclusive connection of human genomes, the shortcoming in adequate representation of that connection is an important flaw in the current state of dimensionality reduction methods improved by LVSDE but not entirely.

LVSDE also limits the number of projected points per data instance to 2 which is simplistic but is not necessarily the best of option for MMDR in terms of possibility of better embeddings.

Lastly, while quantitative evaluation was achieved using a class-aware evaluation measure, any difference between a class-blind evaluation and class-aware evaluation could provide insights for future improvements.

## 10  Conclusion and Future Work

A novel dimensionality reduction method named LVSDE was presented and the philosophy of its underlining platform was discussed and elaborated on. The road ahead is open on many fronts including but not limited to improving speed, and application to data sets in different domains. Even better visual metaphors for visually connecting different projections of a data instance and moving in the direction of preservation of well defined finite point set typology structures may also be a significant research direction in the future. Finding better measures to choose the number of projected points in the red layer is another aspect to improve.

Looking at Figs. 15 and 16, the two configurations of LVSDE that are displayed, while both preferable when compared with t-SNE and UMAP, there is a substantial difference between the two configurations which invites more attention. For the 1000 genomes project data set, on the inclusive connection of human genomes, while configuration 1 of LVSDE is substantially better than UMAP, t-SNE and configuration 2 of LVSDE, there is still room for improvements.

Finally, while LVSDE limits the number of projected points per data instance to 2, going beyond 2 is another direction of interest. Also finding better ways to set the parameters for LVSDE is a further important direction to improve upon. Another interesting direction for improvement is better ways to choose the number of points in the gray layer. While in this paper some quantitative evaluation was reported, it was more to increase confidence in the qualitative visual results rather than as a standalone numerical comparison with careful discussion of the objective, lack of bias and usage scenarios. In particular, in addition to quantitative measures that rely on class numbers, having a proper, suitable and meaningful class number blind quantitative measurement for Multi-Layered Multi-Point Dimensionality Reduction is also of

interest. Such a quantitative measure should be able to take into account multiple neighbourhoods per data instance both in original space and visual space at least to some practical degree.

For ordinary dimensionality reduction, trustworthiness Venna & Kaski (2001); Kaski et al. (2003) has gained attention for matching neighbourhoods in original space and visual space. It is based on a single neighbourhood per data instance both in original space and visual space. While adapting it for multiple neighbourhoods in visual space is less challenging, adapting it for multiple neighbourhoods in original space based on projections of points of original space on a set of hyper lines or a set of manifolds is more challenging and may well prove to be computationally intractable. So practical remedies are desirable on that front as well.

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
