# OpenReview forum: "Multi-point Dimensionality Reduction to Improve Projection Layout Reliability"
_TMLR — Rejected by TMLR_

### Review · Reviewer_Kfg8 · 2023-03-20

**Summary Of Contributions:**

The authors propose an algorithm for multi-point dimensionality reduction. This new task is designed for learning a low-dimensional representation of high-dimensional data that enables the flexibility of embedding a single point into multiple locations in the low-dimensional space. They provide another level of flexibility by splitting points into two layers of representation. The proposed solution, termed Layered Vertex Splitting Data Embedding (LVSDE), builds on a combination of existing dimensionality reduction schemes. The authors apply the new method to several datasets and compare the embedding to the results of UMAP and TSNE.



**Audience:**

Yes

**Broader Impact Concerns:**

No specific ethical concerns.

**Claims And Evidence:**

No

**Requested Changes:**


-The presentation needs dramatic improvement, I am not sure this is feasible as a rebuttal, I think the paper should be completely re written, and the authors should select one main message and focus on it. Currently, it reads like a mix of several ideas.

-Commas missing after many equations.

-Most figures don’t have a clear pointer in the text that explains what is going on in the figure. Referring to 12 figures in one line is not appropriate. For example, the reader can not understand what is going on in figure 1 just based on its caption.

-P3 “the ultimate goal on the application front has been visualization” - I am not sure this statement is correct. DM has numerous other applications besides visualization, such as manifold learning, noise reduction, imputation, clustering, compression, etc.

-Figure 4 - it might be better to visualize where the gray points are located in (a).

-Not sure how you use UMAP to dimension 30? This is not clear; the algorithm is designed for 2d 3d embeddings. Also, once this is applied as a first step, your results are limited by the performance of UMAP.

-Other plots are also not well explained.

-The technical details of the training could be moved to and appendix.


**Strengths And Weaknesses:**

Strengths:

-The proposed problem seems novel and interesting.

-Dimensionality reduction has numerous applications in many fields.

-The concept of using multiple low-dimensional values for a high-dimensional vector is interesting and could lead to follow-up work.

-The English level in the paper is of satisfactory level.

Weakness:
-The paper is poorly written and hard to follow; some parts are completely impossible to understand.

-I don’t completely understand the justification for the multiple points and multiple layers.

-Many elements of the algorithm are not well explained; it is written as a list of elements and modifications with no overview and high-level description or intuition. I seen figure 5, but a text summary would help.

-The experimental results are not convincing, are the numerical results based on multiple runs? Or one run? What are the std? The gap between the proposed method and UMAP or tsne seems marginal and only based on one metric.

-The method is mostly based on the modification of existing schemes, while I can consider this as a contribution, it requires a good systematic evaluation of the contribution of each modification. For example, an ablation study.

-No runtime evaluation of the method, how does it compare to TSNE and UMAP?

-A method termed LDLE was recently introduced by Kohli et al. I believe that it can handle the split issue presented in the paper as a motivating example. Therefore this should be discussed or compared by the authors.


Kohli et al. LDLE: Low Distortion Local Eigenmaps

---

### Review · Reviewer_2XC9 · 2023-03-22

**Summary Of Contributions:**

This work studies the problem of multi-point dimensionality reduction, where each data point in the original high dimensional space can have more than one projection in the 2-dimensional visual space, and the projected points are split into two layers, indicating how reliable each projected point is. To try to solve the problem, an algorithm called Layered Vertex Splitting Data Embedding (LVSDE) is proposed in the paper. The algorithm has four phases, using an existing force-directed graph drawing algorithm to plot a 2d visualization, selecting less reliable projected points and re-computing the 2d visualization with the selected points frozen, reversing frozen points and performing the 2d visualization again, and replicating the less reliable projected points and running the 2d visualization for the last time. Preliminary steps, including UMAP algorithm, distance transformation, and KNN, are required to build a neighborhood graph, which is the input of LVSDE. Numerical experiments on six datasets are performed to show the effectiveness of the proposed algorithm.

**Audience:**

Yes

**Claims And Evidence:**

No

**Requested Changes:**

In my opinion, all comments and questions listed in (1), (2), and (3) in the "Weaknesses" above are required to be addressed, while addressing the comment in (4) would strengthen the work.

**Strengths And Weaknesses:**

Strengths:
(1) The problem of multi-point dimensionality reduction is interesting and can be useful to visualize fuzzy relationship among a dataset.
(2) Some steps in the proposed methods are novel, including the proposed neighborhood-normalized (or density-normalized) distance, the modified forces and masses computation in a force-directed graph drawing algorithm, and the selection and replication of gray (less reliable) projected points via computing the proposed replication pressure.
(3) In numerical experiments, results of LVSDE with different settings of parameters show that the proposed algorithm can have the versatility to adopt to different use cases from semantic visualization to clustering.

Weaknesses:
(1) The "proposed method" in section 4 is described more as an experiment design, rather than a generic algorithm.
- There are too many fixed numbers in the algorithm that should depend on the data dimension or other problem-specific factors. For example, in section 4.2.1 it makes little sense to fix the UMAP projection dimension to 30.
- Is there any mathematical indication that the proposed method could converge to a stable status with forces balanced under certain condition? Is there any justification that the number of iterations, such as 1830, should be fixed instead of problem-specific?
- The parameter setup in section 4.5 is also not generic enough. For example, should the number of neighbors also depend on data dimension?

(2) Solidness and clarity of this work can be improved by better mathematical definition and presentation. Some examples are as follows.
- In Definition 3.2, is $S_i \cup S_j$ required to be empty? It should be noted that sets $S_i$ for all $i$ are fixed and given, not minimization variables.
- Definition 3.3 does not give a formal objective function that multi-point dimensionality reduction tries to minimize. Since this is the core problem that the work tries to solve, it is fundamental that the problem itself can be rigorously defined in math.
- Definition 3.4 and Definition 3.5 are also not in math. It would be better to first mathematically define what reliability of projected point means and then mathematically define (strict) red gray embedding on top of that.
- Computation of replication pressure in section 4.3.3 can be expressed much more concisely and clearly in math.
- Distance from the zth nearest neighborhood of a point $q_i$ should be defined in math.
- In equation (4), what are weight and height?
- In the last paragraph of section 4.3.1, when the temperature first appears, it should be formally defined.
- In Procedure 1, "Bring $s_{it}$ on the frame" should be specified. And at the end of 4.3.5, "slightly bigger" should be more accurate. Otherwise, it is hard to reproduce the proposed algorithm.

(3) Is there any reason to project the forces onto 10 axis when computing the replication pressure in 4.3.3? Why not just computing the net force imposed on each point?

(4) In numerical experiments, it would be better to compared with methods more recently proposed, such as the one in "Agrawal, Akshay, Alnur Ali, and Stephen Boyd. Minimum-distortion embedding. Foundations and Trends® in Machine Learning 14.3 (2021): 211-378", and also other multi-point dimensionality reduction methods.

---

### Review · Reviewer_4ZdN · 2023-03-25

**Summary Of Contributions:**

In this article, the authors provide a method for multi-point dimension reduction, where each point in the original space can have several projections in the lower-dimensional / visual space. For this, (roughly speaking) they first reduce dimension with UMAP, then they compute a nearest-neighbor graph with modified distances, and finally use this graph to compute attractive and repulsive forces on each point / vertex. Based on the values of these forces, points that need to be duplicated are detected (using their so-called replication pressures), and the neighborhood graph is updated with the corresponding copies. This goes on for a fixed number of iterations. Finally, they present a few applications of their method on several common data sets such as MNIST, IRIS and 1000 genomes.

**Audience:**

Yes

**Broader Impact Concerns:**

No concern.

**Claims And Evidence:**

No

**Requested Changes:**

In terms of changes, I think a major rewriting is needed. The paper should be much more concise, and should emphasize the main aspects of the method more while keeping the details separated from the main exposition. I would also like every parameter to be discussed (ie, how to choose it in practice?). This would also allow reader to get a much better grasp of the proposed method. I would also reduce the size of the "philosophy discussion", and only briefly explain why multi-point dimension reduction matters in practice, with only straight to the point arguments and examples. The presentation is currently not so convincing as it is comprised of many vague and hand-waving statements about the field of dimension reduction, which makes that section a bit fishy.

**Strengths And Weaknesses:**

The strengths of this work are listed below.

*** the paper provides a nice adaptation of force-directed graph drawing methods to multi-point dimension reduction. The method is described with a lot of details and seems easy to implement and test.

*** several examples on real data are provided and explained, with complete details on what multi-point dimension reduction brings in addition to usual methods such as t-SNE and UMAP.

The weaknesses of this work are listed below>

*** while I appreciate that the authors provide all necessary details, I think the writing of the article should be improved quite a lot. Currently, it is extremely wordy with a lot of repetitions and typos which actually makes it quite confusing sometimes. It is not always easy to keep a global understanding of the methods as all details (even the unnecessary ones) are presented in front next to the main ideas.

*** the methods has many parameters but most of them are not discussed: why 36 axes in the computation of the replication pressures? why 30 dimensions in the initial UMAP?  why computing normalized distances for the neighborhood graph with arctan function (whereas a squareroot seems also possible based on Fig6)? If normalized distances are used to symmetrize the graph, why not manually adding the edges of opposite direction when they are missing? How is the number of iterations (1830) obtained? Is there any criterion that could be used to detect when one has to stop iterating?

---

### Decision · Action_Editors · 2023-05-11

**Recommendation:** Reject

**Comment:**

The authors did not address reviewers’ concerns. While the problem of multi-point dimensionality reduction is interesting, the paper needs substantial rewriting and modifications.

All three reviewers agreed that major rewriting is needed. The paper should be much more concise and emphasize the main aspects of the method while keeping the details separated from the main exposition. Additionally, each parameter of the proposed method should be discussed in particular regarding how to choose them in practice.


**Audience:**

Since all three reviewers recommended rejection, I would not be able to answer positively to this question.

**Claims And Evidence:**

The claim is that by allowing the points in a low dimensional visual space to be split into two layers – one layer for more reliable points, and another layer for less reliable points we can have a method that outperforms standard dimensionality reduction methods, which mapped each data instance to only one point in a low dimensional space, visually in terms of semantics, group separation, subgroup detection or combinational group detection. This claim is not supported by accurate, convincing and clear evidence. Experimental evidence is not convincing: a) no comparison with more recently proposed, such as the one in "Agrawal, Akshay, Alnur Ali, and Stephen Boyd. Minimum-distortion embedding. Foundations and Trends® in Machine Learning 14.3 (2021): 211-378", and also other multi-point dimensionality reduction methods, b) too many fixed numbers in the algorithm that should depend on the data dimension or other problem-specific factors and there is no ablation study, and c) no repeated evaluations.